# Amortized Inference of Causal Models via Conditional Fixed-Point Iterations

**Divyat Mahajan**[*,1]**, Jannes Gladrow**[2]**, Agrin Hilmkil**[2]**, Cheng Zhang**[2]**, Meyer Scetbon**[*,2]
[1] *Mila, Université de Montréal,* [2] *Microsoft Research*

**Reviewed on OpenReview:** *https://openreview.net/forum?id=D9pq25PGc5*

## Abstract

Structural Causal Models (SCMs) offer a principled framework to reason about interventions and support out-of-distribution generalization, which are key goals in scientific discovery. However, the task of learning SCMs from observed data poses formidable challenges, and often requires training a separate model for each dataset. In this work, we propose an amortized inference framework that trains a single model to predict the causal mechanisms of SCMs conditioned on their observational data and causal graph. We first use a transformer-based architecture for amortized learning of dataset embeddings, and then extend the Fixed-Point Approach (FiP) to infer the causal mechanisms conditionally on their dataset embeddings. As a byproduct, our method can generate observational and interventional data from novel SCMs at inference time, without updating parameters. Empirical results show that our amortized procedure performs on par with baselines trained specifically for each dataset on both in and out-of-distribution problems, and also outperforms them in scarce data regimes .

## 1 Introduction

Learning structural causal models (SCMs) from observations is a core problem in many scientific domains (Sachs et al., 2005; Foster et al., 2011; Xie et al., 2012), as SCMs provide a principled way to model the data generation process. They enable simulation of controlled interventions, offering the potential to accelerate scientific discovery by predicting the outcomes of unseen experiments without requiring costly/time-consuming lab trials (Ke et al., 2023; Zhang et al., 2024). However, solving this inverse problem of learning SCMs from observed data is challenging as both the causal graph and the causal mechanisms are unknown a priori. Recovering causal graphs is an NP-hard combinatorial optimization problem as the space of causal graphs is super-exponential (Chickering et al., 2004). This subsequently complicates the estimation of causal mechanisms via maximum likelihood estimation per node (Blöbaum et al., 2022). To address these challenges, recent approaches have focused on learning causal mechanisms with partial causal structure, using techniques such as autoregressive flows (Khemakhem et al., 2021; Geffner et al., 2022; Javaloy et al., 2023), or modeling SCMs as fixed-point iterations via transformers (Scetbon et al., 2024).

Despite these advances, a major limitation remains: each new dataset requires training a specific model, that prevents the transfer of causal knowledge across datasets. Amortized inference offers a solution by learning a *single* model that can generalize across instances of the same optimization problem by exploiting their shared structure (Andrychowicz et al., 2016; Gordon et al., 2019). This results in models that can quickly adapt to new instances at test time (Finn et al., 2017). Amortized inference has shown success in several challenging tasks, like bayesian posterior estimation (Garnelo et al., 2018; Müller et al., 2021), sampling from unnormalized densities (Akhound-Sadegh et al., 2024; Sendera et al., 2024), as well as causal structure learning (Lorch et al., 2022; Ke et al., 2022), which is more aligned with our paper.

---

[*]Equal Contribution. Correspondence to: `divyatmahajan@gmail.com`.

This work was done when DM was an intern at Microsoft Research. JG, AH, CZ, and MS worked on this project when they were affiliated with Microsoft Research.

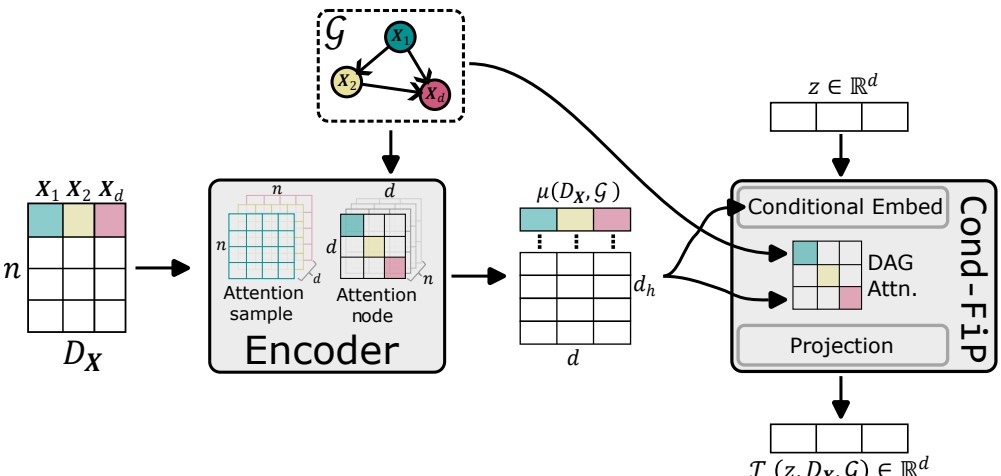

Figure 1: Sketch of the approach proposed in this work. Given a dataset of observations $D_{\boldsymbol{X}}$ and a causal graph $\boldsymbol{\mathcal{G}}$ obtained from an unknown SCM $\mathcal{S}(\mathbb{P}_{\boldsymbol{N}}, \boldsymbol{\mathcal{G}}, \boldsymbol{F})$, the encoder produces a dataset embedding $\mu(D_{\boldsymbol{X}}, \boldsymbol{\mathcal{G}})$, which serves as a condition to instantiate Cond-FiP. Then for any point $\boldsymbol{z} \in \mathbb{R}^d$, $\mathcal{T}(\boldsymbol{z}, D_{\boldsymbol{X}}, \boldsymbol{\mathcal{G}})$ aims at replicating the functional mechanism $\boldsymbol{F}(\boldsymbol{z})$ of the generative SCM.

In this work, we tackle the novel problem of amortized inference of causal mechanisms for additive noise SCMs. While prior research has primarily focused on amortized approaches for causal discovery (Lorch et al., 2022; Dhir et al., 2024) or treatment effect estimation (Nilforoshan et al., 2023; Zhang et al., 2023), our goal instead is to train a single model capable of inferring the causal mechanisms of novel SCMs, given their observational data and associated causal graph. We propose a two-step approach where we first learn dataset embeddings via in-context learning (Garg et al., 2022) to represent the task-specific information. These embeddings are then used to condition the fixed-point (FiP) approach (Scetbon et al., 2024) for modeling causal mechanisms. This conditional modification, termed *Cond-FiP*, enables the model to adapt the causal mechanism for each specific instance (Figure 1). Our key contributions are highlighted below.

- We propose Cond-FiP, a novel extension of FiP approach that enables amortized inference by training a single model across different instances from the functional class of SCMs.

- For novel SCMs at inference, Cond-FiP can recover the causal mechanisms from the input observations without updating any parameters, thereby allowing us to generate observational and interventional data on the fly.

- We show empirically [1] that Cond-FiP achieves similar performances as the state-of-the-art (SOTA) approaches trained from scratch for each dataset on both in and out-of-distribution (OOD) problems. Further, Cond-FiP obtains better results than baselines in scarce data regimes, due to its amortized inference procedure.

## 1.1 Related Works

**Amortized Causal Learning.** Amortized inference has gained traction in causality research, particularly for structure learning. Early works by Lorch et al. (2022) and Ke et al. (2022) introduced transformer-based models trained on multiple synthetic datasets using supervised objectives for amortized inference of causal structure. Their approach aligns with recent works on in-context learning of function classes via transformers (Müller et al., 2021; Akyürek et al., 2022; Garg et al., 2022; Von Oswald et al., 2023). Subsequent improvements targeted OOD generalization (Wu et al., 2024), bayesian causal discovery (Dhir et al., 2024), and applications to gene regulatory networks (Ke et al., 2023; Kim et al., 2025). Beyond structure learning,

---

[1]The code is available on Github: microsoft/causica.

amortized methods have been developed for ATE/CATE estimation (Nilforoshan et al., 2023; Zhang et al., 2023; Sauter et al., 2025; Bynum et al., 2025; Balazadeh et al., 2025; Robertson et al., 2025; Ma et al., 2025; Dhir et al., 2025), model selection (Gupta et al., 2023), and partial causal discovery tasks such as learning topological order (Scetbon et al., 2024). However, amortized inference of causal mechanisms in SCMs remains unaddressed, which is the central focus of our work.

**Autoregressive Causal Learning.** Most causal discovery methods focus first on structure learning (Chickering, 2002; Peters et al., 2014; Zheng et al., 2018), followed by per-node maximum likelihood estimation to recover the causal mechanisms (Blöbaum et al., 2022). In constrast, recent works on causal autoregressive flows (Khemakhem et al., 2021; Geffner et al., 2022; Javaloy et al., 2023) focus on SOTA normalizing flow based generative models to infer causal mechanisms. Further, FiP (Scetbon et al., 2024) modeled SCMs as fixed-point problems over causal (topological) ordering of nodes using transformer-based architectures. These approaches efficiently learn SCMs but require training a separate model per dataset. In this work, we remove this limitation by extending FiP to enable amortized inference of causal mechanisms across different SCMs.

## 2 Amortized Causal Learning

### 2.1 Brief Overview of Amortized Inference

Amortized inference aims to learn a shared inference mechanism across multiple tasks that enables fast adaptation to new tasks at test time. We motivate amortized inference through the following example, consider the task of predicting the motion of objects on different planets. While trajectories vary across planets across due to differences in gravitational constants, the underlying physical laws remain the same. Rather than training a new model from scratch for each planet, we can exploit this shared structure to rapidly adapt our predictions to new settings. This is the core idea behind amortized inference, leveraging common patterns across training tasks to enable fast and efficient adaptation to novel ones.

Consider a task $T$ that defines a distribution over inputs ($\boldsymbol{Z}$) and targets ($\boldsymbol{Y}$); $\boldsymbol{Z}, \boldsymbol{Y} \sim \mathbb{P}_T$. Given a collection of tasks $\left(T^{(k)}\right)_{k=1}^{K}$ and some objective function $L$, the goal is to learn a shared model $\mathcal{T}_\theta$ across tasks:

$$\arg \min_\theta \sum_k \mathbb{E}_{\boldsymbol{Z}, \boldsymbol{Y} \sim \mathbb{P}_{T^{(k)}}} L(\boldsymbol{Y}, \mathcal{T}_\theta(\boldsymbol{Z}, \boldsymbol{I}^{(k)})) \tag{1}$$

where $\boldsymbol{I}^{(k)}$ denotes additional context for task $T^{(k)}$, such as dataset with samples $[\boldsymbol{Z}_1, \cdots, \boldsymbol{Z}_n]$. Instead of retraining from scratch, the model should leverage the context $\boldsymbol{I}'$ to adapt to the task $T'$.

A classic approach towards this is meta-learning (Andrychowicz et al., 2016; Finn et al., 2017; Hospedales et al., 2021), which leverages the context $\boldsymbol{I}'$ by task-specific finetuning. These methods typically learn a shared initialization that is adapted for a new task via few gradient steps in an inner optimization loop. In contrast, in-context learning (ICL) (Müller et al., 2021; Xie et al., 2021; Garg et al., 2022; Akyürek et al., 2024) avoids this inner loop by using transformer-based architectures. By attending to the context $\boldsymbol{I}'$ during the forward pass, ICL methods adapt to a new task without any parameter updates. This capability is often attributed to transformers' ability to implicitly approximate learning algorithms such as gradient descent within their activations (Akyürek et al., 2022; Von Oswald et al., 2023).

While ICL is often used to describe "emergent" test-time adaptation in large language models (Brown et al., 2020), where the training objective does not explicitly involve those tasks. Here, we focus on its formulation as prior-fitted networks (PFNs) (Hollmann et al., 2022; Robertson et al., 2024), where transformers are pretrained on synthetic datasets generated from a simulator that implicitly defines a prior over tasks.

### 2.2 Problem Setup

We start by formally defining structural causal models (SCMs). An SCM defines the causal generative process of a set of $d$ endogenous (causal) random variables $\boldsymbol{V} = \{X_1, \cdots, X_d\}$, where each causal variable $X_i$ is defined as a function of a subset of other causal variables ($\boldsymbol{V} \setminus \{X_i\}$) and an exogenous noise variable $N_i$:

$$X_i = F_i(PA(X_i), N_i) \text{ s.t. } PA(X_i) \subset \boldsymbol{V} , X_i \notin PA(X_i) \tag{2}$$

Hence, an SCM $\mathcal{S}(\mathbb{P}_{\boldsymbol{N}}, \boldsymbol{\mathcal{G}}, \boldsymbol{F})$ describes the data-generation process of $\boldsymbol{X} := [X_1, \cdots, X_d] \sim \mathbb{P}_{\boldsymbol{X}}$ from the noise variables $\boldsymbol{N} := [N_1, \cdots, N_d] \sim \mathbb{P}_{\boldsymbol{N}}$ via the function $\boldsymbol{F} := [F_1, \cdots, F_d]$, and a graph $\boldsymbol{\mathcal{G}} \in \{0,1\}^{d \times d}$ indicating the parents of each $X_i$, that is $[\boldsymbol{\mathcal{G}}]_{i,j} := 1$ if $X_j \in PA(X_i)$. We make the following assumptions about SCMs.

- $\boldsymbol{\mathcal{G}}$ is a directed and acyclic graph (DAG), and noise variables are mutually independent (Markovian SCM).

- SCMs are restricted to be *additive noise models* (ANM), i.e., $X_i = F_i(PA(X_i)) + N_i$.

While the first assumption is pretty standard, we make the ANM assumption for training the proposed dataset encoder in Section 3.1.

Consider a distribution over SCMs $\mathcal{S}(\mathbb{P}_{\boldsymbol{N}}, \boldsymbol{\mathcal{G}}, \boldsymbol{F}) \sim \mathbb{P}_{\mathcal{S}}$. Then the goal with amortized inference of causal mechanisms is to learn a single model $\mathcal{T}_{\theta}$ that can approximate the true causal mechanism $\boldsymbol{F}(\boldsymbol{z})$ for any input $\boldsymbol{z} \in \mathbb{R}^d$. With task specific context as $\boldsymbol{I} = (D_{\mathbf{X}}, \boldsymbol{\mathcal{G}})$ in equation 1, we have

$$\arg\min_{\theta} \mathbb{E}_{\mathcal{S} \sim \mathbb{P}_{\mathcal{S}}} \mathbb{E}_{\boldsymbol{z} \sim \mathbb{P}_{\boldsymbol{X}}} L(\boldsymbol{F}(\boldsymbol{z}), \mathcal{T}_{\theta}(\boldsymbol{z}, D_{\mathbf{X}}, \boldsymbol{\mathcal{G}})) \tag{3}$$

Note that we consider access to causal graph $\boldsymbol{\mathcal{G}}$ as part of the input context, which is available when training on synthetic SCMs. Even if we don't have access to $\boldsymbol{\mathcal{G}}$, we can use prior works on amortized causal learning (Lorch et al., 2022; Ke et al., 2022) to infer the causal graphs from observations $D_{\boldsymbol{X}}$. This justifies our setup where the causal graphs are provided as part of the context to the model.

## 3 Methodology: Conditional FiP

We present our methodology for learning the model $\mathcal{T}(., D_{\boldsymbol{X}}, \boldsymbol{\mathcal{G}})$ that consists of two components: (1) a dataset encoder that generates dataset embeddings $\mu(D_{\boldsymbol{X}}, \boldsymbol{\mathcal{G}})$ from the input context, and (2) a conditional variant of FiP (Scetbon et al., 2024), termed Cond-FiP that allows it to leverage the task-specific context for amortized inference via the learned dataset embeddings $\mu(D_{\boldsymbol{X}}, \boldsymbol{\mathcal{G}})$. We first present our dataset encoder, then Cond-FiP, and conclude with data generation via Cond-Fip.

### 3.1 Dataset Encoder

The objective of this section is to develop a method capable of producing efficient latent representations of datasets. To achieve this, we propose to train an encoder that predicts the noise samples from their associated observations given the causal structures via in-context learning (pseudo code in Algorithm 1, Appendix A.4).

**Training Setting.** We consider empirical representations of $K$ SCMs $\left( \mathcal{S}(\mathbb{P}_{\boldsymbol{N}}^{(k)}, \boldsymbol{\mathcal{G}}^{(k)}, \boldsymbol{F}^{(k)}) \right)_{k=1}^{K}$, each sampled independently from a distribution over SCMs $\mathcal{S}(\mathbb{P}_{\boldsymbol{N}}^{(k)}, \boldsymbol{\mathcal{G}}^{(k)}, \boldsymbol{F}^{(k)}) \sim \mathbb{P}_{\mathcal{S}}$. Each empirical representation, denoted $(D_{\boldsymbol{X}}^{(k)}, \boldsymbol{\mathcal{G}}^{(k)})_{k=1}^{K}$, contains $n$ observations $D_{\boldsymbol{X}}^{(k)} := [\boldsymbol{X}_1^{(k)}, \ldots, \boldsymbol{X}_n^{(k)}]^T \in \mathbb{R}^{n \times d}$, and the causal graph $\boldsymbol{\mathcal{G}}^{(k)} \in \{0,1\}^{d \times d}$. For training, we also need the associated noise samples $D_{\boldsymbol{N}}^{(k)} := [\boldsymbol{N}_1^{(k)}, \ldots, \boldsymbol{N}_n^{(k)}]^T \in \mathbb{R}^{n \times d}$, which play the role of the target variable in our prediction task. For simplicity, we drop the index $k$ in our notation and assume access to the full distribution $\mathbb{P}_{\mathcal{S}}$. The objective is to recover the true noise $D_{\boldsymbol{N}}$ from a dataset of observations $D_{\boldsymbol{X}}$ and the causal graph $\boldsymbol{\mathcal{G}}$, which provide us with dataset embeddings as detailed below.

**Encoder Architecture.** Following (Lorch et al., 2021; Scetbon et al., 2024), we encode datasets using a transformer-based architecture that alternates attention over both sample and node dimension. Given a dataset $D_{\boldsymbol{X}}$, we first apply a linear embedding $L(D_{\boldsymbol{X}}) \in \mathbb{R}^{n \times d \times d_h}$, where $d_h$ is the hidden dimension. The encoder $E$ then applies transformer blocks, each comprising self-attention followed by an MLP (Vaswani et al., 2017), where the attention mechanism is applied either across the samples $n$ or the nodes $d$ alternately. Recall the standard self-attention is defined as

$$\mathrm{A}_{\boldsymbol{M}}(\boldsymbol{Q}, \boldsymbol{K}) = \frac{\exp((\boldsymbol{Q}\boldsymbol{K}^T - \boldsymbol{M})/\sqrt{d_h})}{\exp((\boldsymbol{Q}\boldsymbol{K}^T - \boldsymbol{M})/\sqrt{d_h}) \, \mathbf{1}_d}$$

where $\boldsymbol{Q}, \boldsymbol{K} \in \mathbb{R}^{d \times d_h}$ denote the keys and queries for a single attention head, and $\boldsymbol{M} \in \{0, +\infty\}^{d \times d}$ is a (potential) mask. When attending over samples, the encoder uses standard self-attention without masking ($\boldsymbol{M} = \{0\}^{n \times n}$). But for node-wise attention, we incorporate causal structure by masking invalid dependencies using mask $\boldsymbol{M} = +\infty \times (1 - \boldsymbol{G})$ in standard self-attention, with the convention that $0 \times (+\infty) = 0$. Finally, the embeddings $E(L(D_{\boldsymbol{X}}), \boldsymbol{G}) \in \mathbb{R}^{n \times d \times d_h}$ are passed to a prediction network $H : \mathbb{R}^{n \times d \times d_h} \to \mathbb{R}^{n \times d}$, implemented as 2-hidden layers MLP to project back to the original data space.

**Training Procedure.** We minimize the mean squared error (MSE) of predicting the target $D_{\boldsymbol{N}}$ from the input $(D_{\boldsymbol{X}}, \boldsymbol{G})$ over the distribution of SCMs $\mathbb{P}_{\mathcal{S}}$ available during training:

$$\mathbb{E}_{\mathcal{S} \sim \mathbb{P}_{\mathcal{S}}} \| D_{\boldsymbol{N}} - H \circ E(L(D_{\boldsymbol{X}}), \boldsymbol{G}) \|_2^2 .$$

Further, as we restrict ourselves to the case of ANMs, we can equivalently reformulate our training objective in order to predict the causal mechanism rather than the noise samples, as $\boldsymbol{F}(D_{\boldsymbol{X}}) := D_{\boldsymbol{X}} - D_{\boldsymbol{N}}$. Therefore, we instead propose to train our encoder as follows:

$$\mathbb{E}_{\mathcal{S} \sim \mathbb{P}_{\mathcal{S}}} \| \boldsymbol{F}(D_{\boldsymbol{X}}) - H \circ E(L(D_{\boldsymbol{X}}), \boldsymbol{G}) \|_2^2 .$$

Note that ANM assumption provides a simplified true mapping from data to noise as $x \to x - F(x)$, which is difficult to obtain in general SCMs. Please check Appendix A.2 for more details on justification for ANMs and why recovering noise is equivalent to learning the inverse SCM.

**Inference.** Given a new dataset $D_{\boldsymbol{X}}$ and its causal graph $\boldsymbol{G}$, encoder provides us with the dataset embedding $\mu(D_{\boldsymbol{X}}, \boldsymbol{G}) := E(L(D_{\boldsymbol{X}}), \boldsymbol{G}) \in \mathbb{R}^{n \times d \times d_h}$.

## 3.2 Cond-FiP: Conditional Fixed-Point Decoder

We now present the modification of FiP that uses the learned dataset embeddings $\mu(D_{\boldsymbol{X}}, \boldsymbol{G})$ for amortized inference of causal mechanisms (pseudo code in Algorithm 2, Appendix A.4).

**Training Setting.** Analogous to the encoder training setup, we assume that we have access to a distribution of SCMs $\mathcal{S}(\mathbb{P}_{\boldsymbol{N}}, \boldsymbol{G}, \boldsymbol{F}) \sim \mathbb{P}_{\mathcal{S}}$ at training time, from which we can extract empirical representations $(D_{\boldsymbol{X}}, \boldsymbol{G})$. Our goal is to train $\mathcal{T}$ such that given the context $(D_{\boldsymbol{X}}, \boldsymbol{G})$ from an SCM $\mathcal{S}(\mathbb{P}_{\boldsymbol{N}}, \boldsymbol{G}, \boldsymbol{F}) \sim \mathbb{P}_{\mathcal{S}}$, the induced conditional function $\boldsymbol{z} \in \mathbb{R}^d \to \mathcal{T}(\boldsymbol{z}, D_{\boldsymbol{X}}, \boldsymbol{G}) \in \mathbb{R}^d$ approximates the true causal mechanisms $\boldsymbol{F} : \boldsymbol{z} \in \mathbb{R}^d \to \boldsymbol{F}(\boldsymbol{z}) \in \mathbb{R}^d$ (E.q. 3).

**Decoder Architecture.** The design of our decoder is based on the FiP architecture for fixed-point SCM learning, with two major differences: (1) we use the dataset embeddings $\mu(D_{\boldsymbol{X}}, \boldsymbol{G})$ as a high dimensional codebook to embed the nodes, and (2) we leverage adaptive layer norm operators (Peebles & Xie, 2023) in the transformer blocks of FiP to enable conditional attention mechanisms.

**Conditional Embedding.** The key change of our decoder compared to the original FiP is in the embedding of the input. FiP proposes to embed a data point $\boldsymbol{z} := [z_1, \ldots, z_d] \in \mathbb{R}^d$ into a high dimensional space using a learnable codebook $\boldsymbol{C} := [C_1, \ldots, C_d]^T \in \mathbb{R}^{d \times d_h}$ and positional embedding $\boldsymbol{P} := [P_1, \ldots, P_d]^T \in \mathbb{R}^{d \times d_h}$, from which they define:

$$\boldsymbol{z}_{\text{emb}} := [z_1 * C_1, \ldots, z_d * C_d]^T + \boldsymbol{P} \in \mathbb{R}^{d \times d_h}$$

This ensures that the embedded samples preserve the original causal structure. However, this embedding layer is only adapted if the samples considered are all drawn from the same observational distribution, as the representation of the nodes given by the codebook $\boldsymbol{C}$, is fixed. In order to generalize their embedding strategy to the case where multiple SCMs are considered, we consider conditional codebooks and positional embeddings adapted for each dataset. Given a dataset $D_{\boldsymbol{X}}$ and a causal graph $\boldsymbol{G}$, we propose to define the conditional codebook and positional embedding as

$$\boldsymbol{C}(D_{\boldsymbol{X}}, \boldsymbol{G}) := \mu(D_{\boldsymbol{X}}, \boldsymbol{G}) W_{\boldsymbol{C}}$$
$$\boldsymbol{P}(D_{\boldsymbol{X}}, \boldsymbol{G}) := \mu(D_{\boldsymbol{X}}, \boldsymbol{G}) W_{\boldsymbol{P}}$$

where $\mu(D_{\boldsymbol{X}}, \boldsymbol{\mathcal{G}}) := \text{MaxPool}(E(L(D_{\boldsymbol{X}}), \boldsymbol{\mathcal{G}})) \in \mathbb{R}^{d \times d_h}$ is obtained by max-pooling w.r.t the sample dimension the dataset embedding $E(L(D_{\boldsymbol{X}}), \boldsymbol{\mathcal{G}}) \in \mathbb{R}^{n \times d \times d_h}$ produced by our trained encoder, and $W_{\boldsymbol{C}}, W_{\boldsymbol{P}} \in \mathbb{R}^{d_h \times d_h}$ are learnable parameters. Then we propose to embed any point $\boldsymbol{z} \in \mathbb{R}^d$ conditionally on the context $(D_{\boldsymbol{X}}, \boldsymbol{\mathcal{G}})$ as follows:

$$\boldsymbol{z}_{\text{emb}} := [z_1 * C_1(D_{\boldsymbol{X}}, \boldsymbol{\mathcal{G}}), \dots, z_d * C_d(D_{\boldsymbol{X}}, \boldsymbol{\mathcal{G}})]^T + \boldsymbol{P}(D_{\boldsymbol{X}}, \boldsymbol{\mathcal{G}}) \in \mathbb{R}^{d \times d_h}$$

**Adaptive Transfomer Block.** Once an input $\boldsymbol{z} \in \mathbb{R}^d$ has been embedded as $\boldsymbol{z}_{\text{emb}} \in \mathbb{R}^{d \times d_h}$, FiP models SCMs by simulating the reconstruction of the data from noise. Starting from $\boldsymbol{n}_0 \in \mathbb{R}^{d \times d_h}$ a learnable parameter, they propose to update the current noise $L \geq 1$ times by computing:

$$\boldsymbol{n}_{\ell+1} = h(\text{DA}_{\boldsymbol{M}}(\boldsymbol{n}_\ell, \boldsymbol{z}_{\text{emb}})\boldsymbol{z}_{\text{emb}} + \boldsymbol{n}_\ell)$$

where $h$ refers to the MLP block, and for clarity, we omit both the layer's dependence on its parameters and the inclusion of layer normalization in the notation. Note that here FiP considers the DAG-Attention mechanism (details in Appendix A.1) in order to correctly model the root nodes of the SCM. To obtain a conditional formulation, we first replace the starting noise $\boldsymbol{n}_0$ with $\boldsymbol{n}_0 := \mu(D_{\boldsymbol{X}}, \boldsymbol{\mathcal{G}})W_{\boldsymbol{n}_0} \in \mathbb{R}^{d \times d_h}$, where $W_{\boldsymbol{n}_0} \in \mathbb{R}^{d_h \times d_h}$ is a learnable parameter. Then we add adaptive layer normalization operators (Peebles & Xie, 2023) to both attention and MLP blocks, where each scale or shift is obtained by applying a 1 hidden-layer MLP to the embedding $\mu(D_{\boldsymbol{X}}, \boldsymbol{\mathcal{G}})$.

**Projection.** To project back the latent representation of $\boldsymbol{z}$ obtained from previous stages, $\boldsymbol{n}_L \in \mathbb{R}^{d \times d_h}$, we use a linear operation to get $\widehat{\boldsymbol{z}} = \boldsymbol{n}_L W_{\text{out}} \in \mathbb{R}^d$, where $W_{\text{out}} \in \mathbb{R}^{d_h}$ is learnable.

**Training Procedure.** The result of forward pass can be summarized as $\widehat{\boldsymbol{z}} = \mathcal{T}(\boldsymbol{z}, D_{\boldsymbol{X}}, \boldsymbol{\mathcal{G}})$, where we omit the dependence of $\widehat{\boldsymbol{z}}$ on context $(D_{\boldsymbol{X}}, \boldsymbol{\mathcal{G}})$ for simplicity. We train the model $\mathcal{T}$ by minimizing the reconstruction error of the true causal mechanisms estimated by our model over the distribution of SCMs $\mathbb{P}_{\mathcal{S}}$, as shown below.

$$\mathbb{E}_{\mathcal{S} \sim \mathbb{P}_{\mathcal{S}}} \mathbb{E}_{\boldsymbol{z} \sim \mathbb{P}_{\boldsymbol{X}}} \|\mathcal{T}(\boldsymbol{z}, D_{\boldsymbol{X}}, \boldsymbol{\mathcal{G}}) - \boldsymbol{F}(\boldsymbol{z})\|_2^2 \tag{4}$$

where $\boldsymbol{z} \sim \mathbb{P}_{\boldsymbol{X}}$ is chosen independent of the random dataset $D_{\boldsymbol{X}}$. To compute (4), we propose to sample $n$ independent samples $\boldsymbol{X}_1', \dots, \boldsymbol{X}_n'$ from $\mathbb{P}_{\boldsymbol{X}}$, leading to a new dataset $D_{\boldsymbol{X}'}$ independent of $D_{\boldsymbol{X}}$, and we obtain the following optimzation problem:

$$\mathbb{E}_{\mathcal{S} \sim \mathbb{P}_{\mathcal{S}}} \|\mathcal{T}(D_{\boldsymbol{X}'}, D_{\boldsymbol{X}}, \boldsymbol{\mathcal{G}}) - \boldsymbol{F}(D_{\boldsymbol{X}'})\|_2^2 .$$

**Remark on ANM assumption.** Though our method relies on the ANM assumption for encoder training (Appendix A.2), we do not need this assumption for decoder training! An interesting direction for future work is to develop more general encoder training strategies, such as using self-supervised learning for dataset encoding. Another option is to pursue end-to-end training of Cond-FiP, or adopt curriculum learning by first training the encoder to predict noise variables (as a "pretraining" step), and then fine-tuning it using the decoder's reconstruction loss on more realistic SCMs. However, we consider these extensions beyond the scope of the current work.

## 3.3 Inference with Cond-FiP

We provide a summary of inference procedure with Cond-FiP, with details in Appendix A.3.

**Observational Generation.** Cond-FiP is capable of generating new data samples: given a random vector noise $\boldsymbol{n} \sim \mathbb{P}_{\boldsymbol{N}}$, we can estimate the observational sample associated according to an unknown SCM $\mathcal{S}(\mathbb{P}_{\boldsymbol{N}}, \boldsymbol{\mathcal{G}}, \boldsymbol{F}) \sim \mathbb{P}_{\mathcal{S}}$ as long as we have access to its empirical representation $(D_{\boldsymbol{X}}, \boldsymbol{\mathcal{G}})$. Formally, starting from $\boldsymbol{n}_0 = \boldsymbol{n}$, we infer the associated observation by computing for $\ell = 1, \dots, d$:

$$\boldsymbol{n}_\ell = \mathcal{T}(\boldsymbol{n}_{\ell-1}, D_{\boldsymbol{X}}, \boldsymbol{\mathcal{G}}) + \boldsymbol{n} . \tag{5}$$

After (at most) $d$ iterations, $\boldsymbol{n}_d$ corresponds to the observational sample associated to the original noise $\boldsymbol{n}$ according to our conditional SCM $\mathcal{T}(\cdot, D_{\boldsymbol{X}}, \mathcal{G})$. To sample noise from $\mathbb{P}_{\boldsymbol{N}}$, we leverage cond-FiP that can estimates noise samples under the ANM assumption by computing $\widehat{D_{\boldsymbol{N}}} := D_{\boldsymbol{X}} - \mathcal{T}(D_{\boldsymbol{X}}, \mu(D_{\boldsymbol{X}}, \mathcal{G}))$. From these estimated noise samples, we can efficiently estimate the joint distribution of the noise by computing the inverse cdfs of the marginals as proposed in FiP.

**Interventional Generation.** Cond-FiP also enables the estimation of interventions given an empirical representation $(D_{\boldsymbol{X}}, \mathcal{G})$ of an unkown SCM $\mathcal{S}(\mathbb{P}_{\boldsymbol{N}}, \mathcal{G}, \boldsymbol{F}) \sim \mathbb{P}_{\mathcal{S}}$. To achieve this, we start from a noise sample $\boldsymbol{n}$, and we generate the associated intervened sample $\widehat{\boldsymbol{z}}^{\mathrm{do}}$ by directly modifying the conditional SCM provided by Cond-FiP. More specifically, we modify in place the SCM obtained by Cond-FiP, leading to its interventional version $\mathcal{T}^{\mathrm{do}}(\cdot, D_{\boldsymbol{X}}, \mathcal{G})$. Now, generating an intervened sample can be done by applying the loop defined in (5), starting from $\boldsymbol{n}$ and using the intervened SCM $\mathcal{T}^{\mathrm{do}}(\cdot, D_{\boldsymbol{X}}, \mathcal{G})$ rather than the original one.

# 4 Experiments

## 4.1 Setup

**Data Generation Process.** We use the synthetic data generation procedure proposed by Lorch et al. (2022) to generate SCMs as this framework supports a wide variety of SCMs, making it well-suited for amortized training. It allows sampling of graphs from different schemes and noise variables from diverse distributions. Further, we can also control the complexity of causal mechanisms, choosing between linear (*LIN*) functions or random fourier features (*RFF*) for non-linear causal mechanisms. We construct two distribution of SCMs, $\mathbb{P}_{\mathrm{IN}}$, and $\mathbb{P}_{\mathrm{OUT}}$, which vary based on the choice for sampling causal graphs, noise variables, and causal relationships, see Appendix B.1 for more details.

**Training Datasets.** We randomly sample$\simeq 4e6$ SCMs from the $\mathbb{P}_{\mathrm{IN}}$ distribution, each with $d = 20$ total nodes. From each SCM, we extract the causal graph $\mathcal{G}$ and generate $n_{\mathrm{train}} = 400$ observations to obtain $D_{\boldsymbol{X}}$. This procedure is used to generate training data both the dataset encoder and Cond-FiP, with each epoch containing $\simeq 400$ randomly generated datasets.

**Test Datasets.** We evaluate the model's generalization both in-distribution and out-of-distribution by sampling test datasets from $\mathbb{P}_{\mathrm{IN}}$ and $\mathbb{P}_{\mathrm{OUT}}$, respectively. The test datasets are categorized as follows: LIN **IN** and RFF **IN** where the SCM are sampled from $\mathbb{P}_{\mathrm{IN}}$ with linear and non-linear causal mechanisms respectively. Similarly, we define LIN **OUT** and RFF **OUT** where the SCMs are sampled from $\mathbb{P}_{\mathrm{OUT}}$ instead. For each category, we vary the total nodes $d \in [10, 20, 50, 100]$ and sample 6 or 9 SCMs per $d$, based on the available schemes for sampling the causal graphs (check Appendix B.1 for details). This results in a total of 120 test datasets, supporting a comprehensive evaluation of the methods. For each SCM we generate $n_{\mathrm{test}} = 800$ samples, split equally into task context $D_{\boldsymbol{X}}$ and queries $D_{\boldsymbol{X}'}$ for evaluation. An interesting aspect of our test setup is we assess the model's ability to generalize to larger graphs ($d = 50$, $d = 100$), despite training only with $d = 20$ node graphs.

**Model Architecture.** For both the dataset encoder and cond-FiP, we set the embedding dimension to $d_h = 256$ and the hidden dimension of MLP blocks to 512. Both of our transformer-based models contains 4 attention layers and each attention consists of 8 attention heads. Please check Appendix B.3 for further details and Cond-FiP's memory and compute requirements.

**Baselines.** We compare Cond-FiP against FiP (Scetbon et al., 2024), DECI (Geffner et al., 2022), and DoWhy (Blöbaum et al., 2022). Since the baselines do not have any amortization procedure, they are trained from scratch on each test setting. For a fair comparison with our method, we use the same context set $D_{\boldsymbol{X}}$ with 400 samples to train the baselines, which was used to obtain the dataset embeddings in Cond-FiP. All the methods are then evaluated on the remaining 400 samples in query set $D_{\boldsymbol{X}'}$. Also, we provide the true graph $\mathcal{G}$ to all the baselines to ensure consistency with Cond-FiP.

To avoid potential confusion, we clarify that the notion of distribution shift is defined w.r.t Cond-FiP's training setup. For the baselines, there is no distribution shift as they are trained on the context ($D_{\boldsymbol{X}}$) drawn from the specific test distribution. The most important comparison is with the baseline FiP, as Cond-FiP is

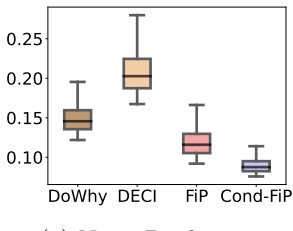
(a) Noise Prediction

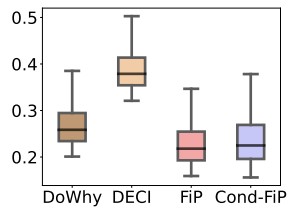
(b) Sample Generation

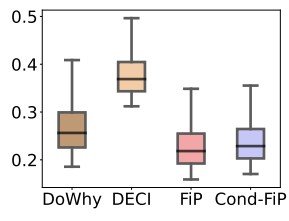
(c) Interventional Generation

Figure 2: **In-Distribution Results.** Benchmarking Cond-FiP for various evaluation tasks, with datasets sampled from RFF **IN** with $d = 20$. The y-axis denotes the RMSE, with mean and standard error over the respective test datasets. Results indicate Cond-FiP can generalize to novel in-distribution instances, with detailed results in Appendix C.

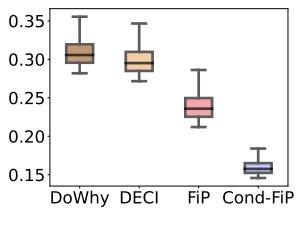
(a) Noise Prediction

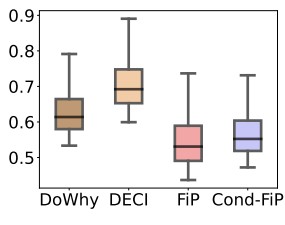
(b) Sample Generation

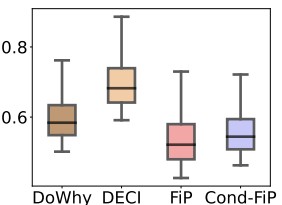
(c) Interventional Generation

Figure 3: **OOD Results.** Benchmarking Cond-FiP for various evaluation tasks, with datasets sampled from RFF **OUT** with $d = 100$ to test for OOD generalization. The y-axis denotes the RMSE, with mean and standard error over the respective test datasets. Results indicate Cond-FiP can generalize to novel OOD instances and larger graphs, with detailed results in Appendix C.

its amortized counterpart. Further, we do not report detailed comparisons with CausalNF (Javaloy et al., 2023) as its performance was consistently weaker than other baselines, check Appendix H for details.

**Evaluation Tasks.** We evaluate the methods on the following three tasks. *Noise Prediction:* given the observations $D_{\boldsymbol{X}}$ and the true graph $\boldsymbol{\mathcal{G}}$, infer the noise variables $\widehat{D_{\boldsymbol{N}}}$. *Sample Generation:* given the noise samples $D_{\boldsymbol{N}}$ and the true graph $\boldsymbol{\mathcal{G}}$, generate the causal variables $\widehat{D_{\boldsymbol{X}}}$. *Interventional Generation:* generate intervened samples from noise samples $D_{\boldsymbol{N}}$ and the true graph $\boldsymbol{\mathcal{G}}$.

**Metric.** Let us denote a predicted & true target as $\widehat{\boldsymbol{Y}} \in \mathbb{R}^{n_{\text{test}} \times d}$ and $\boldsymbol{Y} \in \mathbb{R}^{n_{\text{test}} \times d}$. Then RMSE is computed as $\frac{1}{n_{\text{test}}} \sum_{i=1}^{n_{\text{test}}} \sqrt{\frac{1}{d} \|[\boldsymbol{Y}]_i - [\widehat{\boldsymbol{Y}}]_i\|_2^2}$. Note that we scale RMSE by dimension $d$, which allows us to compare results across different graph sizes.

## 4.2 Results

**Generalization to OOD data and larger graphs.** In Figure 2, we first present results for in-distribution generalization using test datasets sampled from RFF **IN** for graphs with $d = 20$ nodes. Cond-FiP performs competitively with baselines trained from scratch on each test instance, hence it successfully generalizes to novel in-distribution instances. Notably, Cond-FiP was never explicitly trained to generate interventional data, and its strong performance on this task further supports that it captures the underlying causal mechanisms.

Next we consider the more challenging case of OOD generalization using test datasets sampled from RFF **OUT** and graphs with $d = 100$ nodes, while the Cond-FiP was trained only with $d = 20$ node graphs. As shown in Figure 3, Cond-FiP continues to perform well, indicating successful generalization to OOD instances and significantly larger graphs! Due to space constraints, we report results for SCMs with non-linear mechanisms—the more challenging setting. Full results for both in-distribution and OOD scenarios are available in Appendix C, where our findings remain consistent.

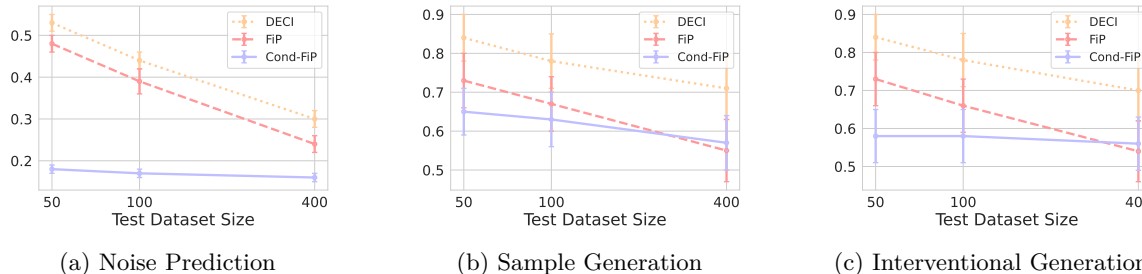

(a) Noise Prediction      (b) Sample Generation      (c) Interventional Generation

Figure 4: **Scarce Data Regime Results.** Benchmarking Cond-FiP on the various evaluation tasks (RFF **OUT** and $d = 100$) as we reduce the test dataset size. The y-axis denotes the RMSE, with mean and standard error over the respective test datasets. Cond-FiP generalizes much better than the baselines in the low-data regime, with detailed results in Appendix E.

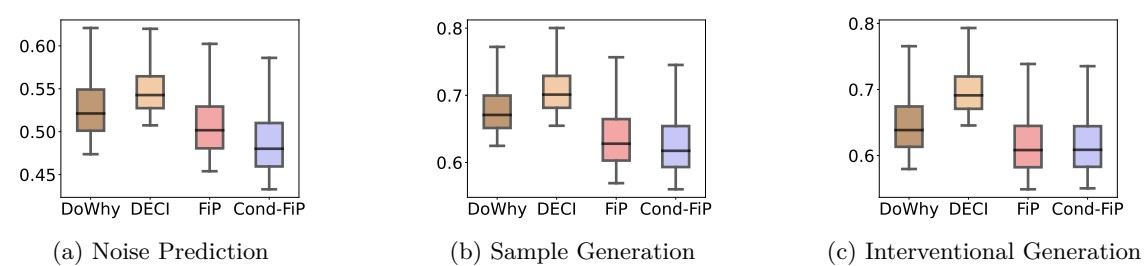

(a) Noise Prediction      (b) Sample Generation      (c) Interventional Generation

Figure 5: **OOD Results without True Graph.** Benchmarking Cond-FiP for various evaluation tasks, with datasets sampled from RFF **OUT** with $d = 100$ where the true graph $\mathcal{G}$ is not present in input context, rather its inferred via AVICI. The y-axis denotes the RMSE, with mean and standard error over the respective test datasets. Results indicate Cond-FiP can generalize to novel instances even in the absence of true graph, with detailed results in Appendix F.

We also assess Cond-FiP's sensitivity to distribution shifts by varying the magnitude of distribution shift (details in Appendix D). We consider two cases, where we control the severity in distribution shift by controlling the causal mechanisms or the noise variables. We find that Cond-FiP is more robust to shifts in causal mechanisms, with minimal performance degradation. However, its performance is more sensitive to shifts in noise distributions, deteriorating as the magnitude of shift increases.

**Better Generalization in Scarce Data Regimes.** An advantage of amortized inference methods is their ability to generalize well when context $D_{\boldsymbol{X}}$ for test instances is small. As the context size decreases, baselines often suffer significant performance drops as they require training from scratch. In contrast, Cond-FiP is less impacted as its parameters remain unchanged at inference time, and the inductive bias learned during training enables effective generalization even with limited context. In Figure 4, we demonstrate this in the challenging OOD setting (RFF **OUT**, $d = 100$), where Cond-FiP outperforms the baselines. Please check Appendix E for further details.

**Generalization without True Causal Graph.** So far, our results assume access to the true causal graph ($\mathcal{G}$) as part of the input context to Cond-FiP. However, Cond-FiP can be extended to operate without this information by first inferring the graph using amortized structure learning methods (Lorch et al., 2022; Ke et al., 2022). We demonstrate this in Figure 5 for the RFF **OUT**; setting with $d = 100$ nodes, using graphs inferred via AVICI (Lorch et al., 2022) for both Cond-FiP and the baselines. The results show that Cond-FiP remains competitive, supporting its ability to capture underlying causal mechanisms (details in Appendix F).

Further, to assess whether Cond-FiP genuinely learns causal mechanisms tied to the input graph structure, we perform a systematic sensitivity analysis by perturbing the causal graph. Starting from the true graph, we randomly remove a proportion ($p$) of the true edges, such that on average $p\times$ (total edges) are missing.

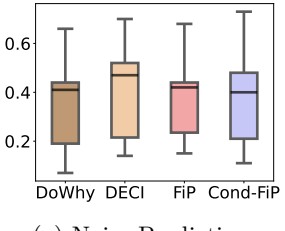 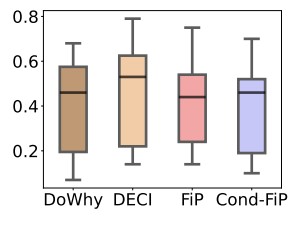 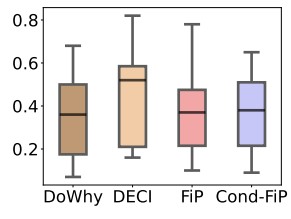

(a) Noise Prediction                (b) Sample Generation                (c) Interventional Generation

Figure 6: **CSuite Results.** Benchmarking Cond-FiP on the various evaluation tasks on the CSuite benchmark, which uses a different data simulator than the Cond-FiP's training data simulator. The y-axis denotes the RMSE, with mean and standard error across the 9 test datasets.

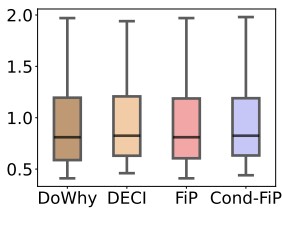 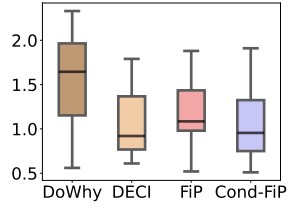 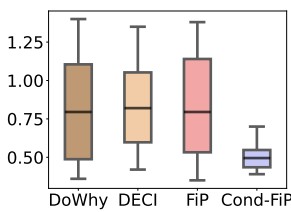

(a) Noise Prediction                (b) Sample Generation                (c) Interventional Generation

Figure 7: **CSuite GMM Results.** Benchmarking Cond-FiP on the Large Backdoor and Weak Arrow datasets from the CSuite benchmark, where the noise distribution is modified to be a multi-modal gaussian mixture model. The y-axis denotes the RMSE, with mean and standard error across the 12 test scenarios. Results indicate that Cond-FiP can generalize to instances with more complex noise distributions like GMMs.

As shown in Table 21 (Appendix F.1), Cond-FiP remains robust under moderate graph perturbations and performs competitively with FiP, which is retrained from scratch for each setting. These results further support that Cond-FiP learns transferable causal mechanisms and can adapt at test time by inferring functions consistent with the available (and potentially imperfect) graph and observational context.

**Ablation Study.** Since the method relies on large-scale pretraining over synthetic SCMs, we analyze how Cond-FiP's performance scales with the number of pretraining SCMs. We conduct experiments at smaller scales, with a total of $1e5, 4e5, 1e6$ SCMs, as opposed to using $4e6$ SCMs in our main results above. Our results in Table 22 (Appendix G.1) show that Cond-FiP benefits consistently from additional pretraining data, though the returns gradually diminish as scale increases. The most pronounced gains appear when increasing the pretraining size from $1e5$ to $4e5$, while improvements become more incremental beyond $1e6$.

Further, we conduct ablation studies on both the encoder (Appendix G.2) and decoder (Appendix G.3) to better understand how the training data affects generalization performance. We find that Cond-FiP remains competitive even when the encoder is trained on only RFF data, compared to training on a mixture of both. In contrast, decoder performance benefits more noticeably from training on the combined dataset.

**Generalization to novel data simulators.** We further evaluate Cond-FiP on test datasets generated using C-Suite (Geffner et al., 2022), a synthetic data simulator distinct from the training simulator. As shown in Figure 6, Cond-FiP generalizes well to these novel instances. Additionally, to conduct more OOD evaluations, we modify the noise distribution of the Large Backdoor and Weak Arrow datasets from the Csuite benchmark such that the noise variables are sampled from a gaussian mixture model (GMM) (details in Appendix B.2). Results in Figure 7 demonstrate that Cond-FiP can generalize to more complex noise distributions as well. Importantly, while baselines were trained from scratch for each specific gaussian mixture noise distribution, Cond-FiP was pretrained only on gaussian noise and generalizes effectively to settings with GMM noise distribution.

| Method | MMD($\widehat{D_{\boldsymbol{X}}^{\text{query}}}, D_{\boldsymbol{X}}^{\text{query}}$) | MMD($\widehat{D_{\boldsymbol{X}}^{\text{context}}}, D_{\boldsymbol{X}}^{\text{query}}$) | MMD($D_{\boldsymbol{X}}^{\text{context}}, D_{\boldsymbol{X}}^{\text{query}}$) |
|---|---|---|---|
| DoWhy | 0.015 | 0.014 | 0.005 |
| DECI | 0.014 | 0.005 | 0.005 |
| FiP | 0.015 | 0.005 | 0.005 |
| Cond-FiP | 0.013 | 0.005 | 0.005 |

Table 1: **Results for Flow Cytometry (Sachs) dataset.** We benchmark Cond-FiP against the baselines for the task of generating observational data on the real world Sachs benchmark. Each cell reports the MMD, and we also report the reconstruction error for all of the methods. *Results indicate that Cond-FiP matches the performance of baselines trained from scratch.*

| Method | MMD($\widehat{D_{\boldsymbol{X}}^{\text{query}}}, D_{\boldsymbol{X}}^{\text{query}}$) | MMD($\widehat{D_{\boldsymbol{X}}^{\text{context}}}, D_{\boldsymbol{X}}^{\text{query}}$) | MMD($D_{\boldsymbol{X}}^{\text{context}}, D_{\boldsymbol{X}}^{\text{query}}$) |
|---|---|---|---|
| DoWhy | 0.020 | 0.014 | 0.005 |
| DECI | 0.016 | 0.005 | 0.005 |
| FiP | 0.017 | 0.005 | 0.005 |
| Cond-FiP | 0.019 | 0.005 | 0.005 |

Table 2: **Results for Ecoli dataset.** We benchmark Cond-FiP against the baselines for the task of generating observational data on the real world Ecoli benchmark from the bnlearn repository. Each cell reports the MMD, and we also report the reconstruction error for all of the methods. *Results indicate that Cond-FiP matches the performance of baselines trained from scratch.*

**Experiments on real-world benchmarks.** Finally, we show that Cond-FiP can generalize to the real-world instances using the flow cytometry dataset (Sachs et al., 2005) and ecoli dataset (Scutari, 2010). Although Cond-FiP cannot be trained on real-world datasets since the encoder requires access to true noise variables, it can still be used for inference. Both datasets contains $n \simeq 800$ observational samples expressed in a $d = 11$ dimensional space for the flow cytometry dataset and $d = 46$ dimensional space for the ecoli dataset, and the corresponding reference (true) causal graph. We split this into context $D_{\boldsymbol{X}}^{\text{context}} \in \mathbb{R}^{n_{\text{context}} \times d}$ and queries $D_{\boldsymbol{X}}^{\text{query}} \in \mathbb{R}^{n_{\text{query}} \times d}$, each of size $n_{\text{context}} = n_{\text{query}} = 400$. Note that the context dataset is to used to train the baselines and obtain dataset embedding for Cond-FiP, while the query dataset is used for evaluation of all the methods.

Since we don't have access to the true causal mechanisms, we cannot compute RMSE for noise prediction or sample generation like we did in our experiments with synthetic benchmarks. Instead for each method, we obtain the noise predictions $\widehat{D_{\boldsymbol{N}}^{\text{context}}}$ on the context, and use it to fit a gaussian distribution for each component (node). Then we use the learned gaussian distribution to sample new noise variables, $\widehat{D_{\boldsymbol{N}}^{\text{query}}}$, which are mapped to the observations as per the causal mechanisms learned by each method, $\widehat{D_{\boldsymbol{X}}^{\text{query}}}$. Finally, we compute the maximum mean discrepancy (MMD) distance between $\widehat{D_{\boldsymbol{X}}^{\text{query}}}$ and $D_{\boldsymbol{X}}^{\text{query}}$ as metric to determine whether the method has captured the true causal mechanisms. For consistency, we also evaluate the reconstruction performances by directly using the inferred noise from context $\widehat{D_{\boldsymbol{N}}^{\text{context}}}$ from the models, and then compute MMD between their reconstructed data ($\widehat{D_{\boldsymbol{X}}^{\text{context}}}$) and the query data ($D_{\boldsymbol{X}}^{\text{query}}$).

Table 1, 2, presents our results for the flow cytometry and the ecoli dataset, where for reference we also report the MMD distance between samples from the context and query split, which should serve as the gold standard since both the datasets are sampled from the same distribution. We find that Cond-FiP is competitive with the baselines that were trained from scratch. Except DoWhy, the MMD distance with reconstructed samples from the methods are close to oracle performance.

Note that Cond-FiP (as well as the other baselines) only supports hard interventions while the interventional data available for Sachs are soft interventions. Hence, we are unable to provide a comprehensive evaluation of Cond-FiP (and the baselines) for interventional predictions on Sachs.

**Computational Efficiency.** Like other amortized approaches, Cond-FiP has a higher training cost than the baselines, as it is trained across multiple datasets. While the cost of each forward-pass is comparable to FiP, we trained Cond-FiP over approximately 4M datasets in an amortized manner. However, Cond-FiP offers a significant advantage at inference time since it requires only a single forward pass to generate predictions, whereas the baselines must be retrained from scratch for each new dataset. Thus, while Cond-FiP incurs a higher one-time training cost, its substantially faster at inference.

For instance, Cond-FiP can infer causal mechanisms for a novel task in under one minute, while FiP takes on average 30 minutes per task. For a concrete comparison, it took us 30 hours to train Cond-FiP but we can solve each inference task in max 1 minute. Therefore, to compute our main results ( Appendix C we evaluated 360 different tasks, implying a total cost of $30 + 360/60 = 36$ hours with Cond-FiP. In contrast, evaluating FiP on the same 360 tasks would require retraining from scratch each time, taking approximately 180 hours!

Thus, while Cond-FiP has a higher one-time training cost, it offers a $5\times$ speedup over FiP in total runtime when evaluating across multiple tasks.

## 5 Conclusion

In this work, we propose novel methodology for training a *single* model for amortized inference of causal mechanisms in SCMs. Cond-FiP not only generalizes to unseen in-distribution instances, but also to a wide range of OOD instances, including larger graphs, unknown causal graphs, complex noise distributions, and real-world data. To the best of our knowledge, this is the first approach to demonstrate the feasibility of learning causal mechanisms in a reusable, foundational manner—paving the way for a paradigmatic shift towards the assimilation of causal knowledge across datasets.

**Limitations.** Our training is limited to synthetic additive noise SCMs due to the requirement of true noise variables for learning the dataset encoder. However, the conditional FiP decoder (see Section 3.2) does not rely on this assumption and can be applied to general SCMs given pretrained dataset embeddings. A promising direction for future work is to explore more general encoding schemes, such as self-supervised learning, or design an implicit in-context learning approach to remove the need for dataset embeddings via direct attention over the context (Mittal et al., 2024). We believe our framework can serve as a good motivation for future works that incorporate real-world datasets during training as well.

While Cond-FiP generalizes to larger graphs, it does not yet benefit from larger context sizes at inference (Appendix I.1), suggesting the need to scale both the model and training data for richer contexts. Additionally, although Cond-FiP performs well on generating interventional samples, it doesn't perform well on counterfactual generation (Appendix I.2). Future work will explore scaling Cond-FiP to larger problem instances and application for more complex tasks (counterfactual generation) in real-world scenarios.

## Acknowledgements

We thank the members of the Machine Intelligence team at Microsoft Research for helpful discussions. We also thank Sarthak Mittal for their suggestion to benchmark Cond-FiP in the scarce data regime. Further, we thank Moksh Jain for their feedback on the draft. Part of the experiments were enabled by the Digital Research Alliance of Canada (`https://alliancecan.ca/en`) and Mila cluster (`https://docs.mila.quebec/index.html`). Divyat Mahajan acknowledges support via FRQNT doctoral scholarship (`https://doi.org/10.69777/354785`) for his graduate studies.

## Broader Impact Statement

We propose novel methodology for amortized inference of causal mechanisms in structural causal models, representing an initial step toward the development of causal foundational models. Integrating causal principles into machine learning has been widely suggested to improve robustness and reliability, an important property for high-stakes domains such as healthcare, policy, and scientific discovery. By advancing core methodology in causal inference, our work may indirectly support the creation of machine learning systems that are more transparent and trustworthy. However, our research currently does not target any societal application, and does not pose foreseeable risks or negative consequences.

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

# Appendix

## Table of Contents

# A  Additional Details on Cond-FiP

## A.1  DAG-Attention Mechanism

In FiP (Scetbon et al., 2024) the authors propose to leverage the transformer architecture to learn SCMs from observations. By reparameterizing an SCM according to a topological ordering induced by its graph, the authors show that any SCM can be reformulated as a fixed-point problem of the form $\boldsymbol{X} = \boldsymbol{H}(\boldsymbol{X}, \boldsymbol{N})$ where $\boldsymbol{H}$ admits a simple triangular structure:

$$[\mathrm{Jac}_{\boldsymbol{x}}\boldsymbol{H}(\boldsymbol{x},\boldsymbol{n})]_{i,j} = 0, \quad \text{if} \quad j \geq i$$
$$[\mathrm{Jac}_{\boldsymbol{n}}\boldsymbol{H}(\boldsymbol{x},\boldsymbol{n})]_{i,j} = 0, \quad \text{if} \quad i \neq j,$$

where $\mathrm{Jac}_{\boldsymbol{x}}\boldsymbol{H}$, $\mathrm{Jac}_{\boldsymbol{n}}\boldsymbol{H}$ denote the Jacobian of $\boldsymbol{H}$ w.r.t the first and second variables respectively. Motivated by this fixed-point reformulation, FiP considers a transformer-based architecture to model the functional relationships of SCMs and propose a new attention mechanism to represent DAGs in a differentiable manner. Recall that the standard attention matrix is defined as:

$$\mathrm{A}_{\boldsymbol{M}}(\boldsymbol{Q}, \boldsymbol{K}) = \frac{\exp((\boldsymbol{Q}\boldsymbol{K}^T - \boldsymbol{M})/\sqrt{d_h})}{\exp((\boldsymbol{Q}\boldsymbol{K}^T - \boldsymbol{M})/\sqrt{d_h})\,\mathbf{1}_d} \tag{6}$$

where $\boldsymbol{Q}, \boldsymbol{K} \in \mathbb{R}^{d \times d_h}$ denote the keys and queries for a single attention head, and $\boldsymbol{M} \in \{0, +\infty\}^{d \times d}$ is a (potential) mask. When M is chosen to be a triangular mask, the attention mechanism (6) enables to parameterize the effects of previous nodes on the current one However, the normalization inherent to the softmax operator in standard attention mechanisms prevents effective modeling of root nodes, which *should not* be influenced by any other node in the graph. To alleviate this issue, FiP proposes to consider the following formulation instead:

$$\mathrm{DA}_{\boldsymbol{M}}(\boldsymbol{Q}, \boldsymbol{K}) = \frac{\exp((\boldsymbol{Q}\boldsymbol{K}^T - \boldsymbol{M})/\sqrt{d_h})}{\mathcal{V}\big(\exp((\boldsymbol{Q}\boldsymbol{K}^T - \boldsymbol{M})/\sqrt{d_h})\,\mathbf{1}_d\big)} \tag{7}$$

where $\mathcal{V}_i(\boldsymbol{v}) = v_i$ if $v_i \geq 1$, else $\mathcal{V}_i(\boldsymbol{v}) = 1$ for any $\boldsymbol{v} \in \mathbb{R}^d$. While softmax forces the coefficients along each row of the attention matrix to sum to one, the attention mechanism described in (7) allows the rows to sum in $[0, 1]$, thus enabling to model root nodes in attention.

## A.2  Details on Encoder Training

**Additive Noise Model Assumption.**  Our method relies on the ANM assumption only for the training the encoder. This is because we require the encoder to predict the noise from data in order to obtain embeddings, and under the ANM assumption, the mapping from data to noise can be easily expressed as $x \to x - F(x)$ where $F$ is the generative functional mechanism of the generative ANM. However, if we were to consider general SCMs, i.e. of the form $X = F(X, N)$, we would need access to the mapping $x \to F^{-1}(x, \cdot)(x)$ (assuming this function is invertible), which for general functions is not tractable. Also, note that the ANM assumption by default ensures invertibility since the jacobian w.r.t noise is a triangular matrix with nonzero diagonal elements. An interesting future work would be to consider a more general dataset encoding (using self-supervised techniques) that do not require the ANM assumption, but we believe this is out of the scope of this work.

We now provide further details on training the encoder and show how recovering the noise is equivalent to learn the inverse causal generative process. Recall that an SCM is an *implicit* generative model that, given a noise sample $\mathbf{N}$, generates the corresponding observation according to the following fixed-point equation in $\mathbf{X}$

$$\mathbf{X} = F(\mathbf{X}, \mathbf{N})$$

More precisely, to generate the associated observation, one must solve the above fixed-point equation in $\mathbf{X}$ given the noise $\mathbf{N}$. Let us now introduce the following notation that will be instrumental for the subsequent discussion: we denote $F_{\mathbf{N}}(z) : z \to F(z, \mathbf{N})$.

Due to the specific structure of $F$ (determined by the DAG $\mathcal{G}$ associated with the SCM), the fixed-point equation mentioned above can be efficiently solved by iteratively applying the function $F_\mathbf{N}$ to the noise (see Eq. (5) in the manuscript). As a direct consequence, the observation $\mathbf{X}$ can be expressed as a function of the noise:

$$\mathbf{X} = F_{\text{gen}}(\mathbf{N})$$

where $F_{\text{gen}}(\mathbf{N}) := (F_\mathbf{N})^{\circ d}(\mathbf{N})$, $d$ is the number of nodes, and $\circ$ denotes the composition operation. In the following we refer to $F_{\text{gen}}$ as the *explicit* generative model induced by the SCM.

Conversely, assuming that the mapping $z \to F_{\text{gen}}(z)$ is invertible, then one can express the noise as a function of the data:

$$\mathbf{N} = F_{\text{gen}}^{-1}(\mathbf{X})$$

Therefore, learning to recover the noise from observation is equivalent to learn the function $F_{\text{gen}}^{-1}$, which is exactly the inverse of the explicit generative model $F_{\text{gen}}$. It is also worth noting that under the ANM assumption (i.e. $F(\mathbf{X}, \mathbf{N}) = f(\mathbf{X}) + \mathbf{N}$), $F_{\text{gen}}$ is in fact always invertible and its inverse admits a simple expression which is

$$F_{\text{gen}}^{-1}(z) = z - f(z)$$

Therefore, in this specific case, learning the inverse generative model $F_{\text{gen}}^{-1}$ is exactly equivalent to learning the causal mechanism function $f$.

### A.3    Inference with Cond-FiP

**Sample Generation.**    Given a dataset $D_\mathbf{X}$ and its causal graph $\mathcal{G}$, we denote $z \to \mathcal{T}(z, D_\mathbf{X}, \mathcal{G})$ the function infered by Cond-FiP. This function defines the predicted SCM obtained by our model, and we can directly use it to generate new points. More precisely, given a noise sample $\mathbf{n}$, we can generate the associated observational sample by solving the following equation in $\mathbf{x}$:

$$\mathbf{x} = \mathcal{T}(\mathbf{x}, D_\mathbf{X}, \mathcal{G}) + \mathbf{n}$$

To solve this fixed-point equation, we rely on the fact that $\mathcal{G}$ is a DAG, which enables to solve the fixed-point problem using the following simple iterative procedure. Starting with $\mathbf{z}_0 = \mathbf{n}$, we compute for $\ell = 1, \ldots, d$ where $d$ is the number of nodes

$$\mathbf{z}_\ell = \mathcal{T}(\mathbf{z}_{\ell-1}, D_\mathbf{X}, \mathcal{G}) + \mathbf{n}$$

After $d$ iterations we obtain the following,

$$\mathbf{z}_d = \mathcal{T}(\mathbf{z}_d, D_\mathbf{X}, \mathcal{G}) + \mathbf{n}$$

Therefore, $\mathbf{z}_d$ is the solution of the fixed-point problem above, which corresponds to the observational sample associated to $\mathbf{n}$ according to our predicted SCM $z \to \mathcal{T}(z, D_\mathbf{X}, \mathcal{G})$.

**Interventional Prediction.**    Recall that given a dataset $D_\mathbf{X}$ and its causal graph $\mathcal{G}$, $z \in \mathbb{R}^d \to \mathcal{T}(z, D_\mathbf{X}, \mathcal{G}) \in \mathbb{R}^d$ denotes the SCM infered by Cond-FiP. Let us also denote the coordinate-wise formulation of our SCM defined for any $z \in \mathbb{R}^d$ as $\mathcal{T}(z, D_\mathbf{X}, \mathcal{G}) = [[\mathcal{T}(z, D_\mathbf{X}, \mathcal{G})]_1, \ldots, [\mathcal{T}(z, D_\mathbf{X}, \mathcal{G})]_d]$, where for all $i \in \{1, \ldots, d\}$, $z \in \mathbb{R}^d \to [\mathcal{T}(z, D_\mathbf{X}, \mathcal{G})]_i \in \mathbb{R}$ is a real-valued function.

In order to intervene on this predicted SCM, we simply have to modify in place the predicted function. For example, assume that we want to perform the following intervention $\text{do}(X_i) = a$. Then, to obtain the intervened SCM, we define a new function $z \to \mathcal{T}^{\text{do}(X_i)=a}(z, D_\mathbf{X}, \mathcal{G})$ defined for any $z \in \mathbb{R}^d$ as: $[\mathcal{T}^{\text{do}(X_i)=a}(z, D_\mathbf{X}, \mathcal{G})]_j := [\mathcal{T}(z, D_\mathbf{X}, \mathcal{G})]_j$ if $j \neq i$ and $[\mathcal{T}^{\text{do}(X_i)=a}(z, D_\mathbf{X}, \mathcal{G})]_i := a$.

Now, using this intervened SCM $z \to \mathcal{T}^{\text{do}(X_i)=a}(z, D_\mathbf{X}, \mathcal{G})$, we can apply the exact same generation procedure as the one introduced above to generate intervened samples according to our intervened SCM.

### A.4 Pseduo Code

---

**Algorithm 1** Cond-FiP Part 1: Dataset Encoder $\mu(D_{\boldsymbol{X}}, \boldsymbol{\mathcal{G}})$

---

**Input:** Observational dataset $D_{\boldsymbol{X}} \in \mathbb{R}^{n \times d}$, causal graph $\boldsymbol{\mathcal{G}} \in \{0, 1\}^{d \times d}$
**Output:** Dataset embedding $\mu(D_{\boldsymbol{X}}, \boldsymbol{\mathcal{G}})$

1: **Linear Embedding:** Apply a learned linear map $L : \mathbb{R}^1 \to \mathbb{R}^{d_h}$ to each sample:

$$L(D_{\boldsymbol{X}}) = [\boldsymbol{X}_1 W_L, \dots, \boldsymbol{X}_n W_L]^\top \in \mathbb{R}^{n \times d \times d_h}.$$

2: **Alternating Attention:** Pass $L(D_{\boldsymbol{X}})$ through a stack of transformer blocks that alternate between:
- *Sample-wise attention* (unmasked) across the $n$ observations, capturing global context.
- *Node-wise attention* with mask defined as per the causal graph.

$$M_{ij} = \begin{cases} 0, & \text{if } G_{ij} = 1, \\ -\infty, & \text{otherwise.} \end{cases}$$

3: **Contextual Representation:** After alternating attention blocks, obtain dataset embeddings

$$\mu(D_{\boldsymbol{X}}, \boldsymbol{\mathcal{G}}) = E(L(D_{\boldsymbol{X}}), \boldsymbol{\mathcal{G}}) \in \mathbb{R}^{n \times d \times d_h}.$$

4: **Prediction Head:** Apply a two-layer MLP $H$ to map embeddings to estimated noise or function values:

$$\hat{Y} = H(\mu(D_{\boldsymbol{X}}, \boldsymbol{\mathcal{G}})) \in \mathbb{R}^{n \times d}.$$

5: **Training Objective:** Under the additive noise model assumption, minimize the MSE loss:

$$\mathcal{L}_E = \mathbb{E}_{S \sim P_S} \left[ \|\boldsymbol{F}(D_{\boldsymbol{X}}) - \hat{Y}\|_2^2 \right].$$

6: **Output:** Return the dataset embedding $\mu(D_{\boldsymbol{X}}, \boldsymbol{\mathcal{G}})$

---

---

**Algorithm 2** Cond-FiP Part 2: Conditional Fixed-Point Decoder $\mathcal{T}(\boldsymbol{z}, D_{\boldsymbol{X}}, \boldsymbol{\mathcal{G}})$

---

**Input:** Input point $\boldsymbol{z} \in \mathbb{R}^d$, dataset $D_{\boldsymbol{X}} \in \mathbb{R}^{n \times d}$, causal graph $\boldsymbol{\mathcal{G}} \in \{0,1\}^{d \times d}$, encoder output $\mu(D_{\boldsymbol{X}}, \boldsymbol{\mathcal{G}})$
**Output:** Learned causal model: $\mathcal{T}(z, D_{\boldsymbol{X}}, \boldsymbol{\mathcal{G}})$
 1: **Pooling:** Aggregate the encoder representations across samples:

$$\mu'(D_{\boldsymbol{X}}, \boldsymbol{\mathcal{G}}) = \mathrm{MaxPool}_n\big(E(L(D_{\boldsymbol{X}}), \boldsymbol{\mathcal{G}})\big) \in \mathbb{R}^{d \times d_h}.$$

 2: **Context-Specific Codebooks:** Compute dataset-conditioned codebooks and positional embeddings:

$$C(D_{\boldsymbol{X}}, \boldsymbol{\mathcal{G}}) = \mu'(D_{\boldsymbol{X}}, \boldsymbol{\mathcal{G}}) W_C, \quad P(D_{\boldsymbol{X}}, \boldsymbol{\mathcal{G}}) = \mu'(D_{\boldsymbol{X}}, \boldsymbol{\mathcal{G}}) W_P.$$

 3: **Input Embedding:** Embed the input $\boldsymbol{z}$ via element-wise modulation:

$$\boldsymbol{z}_{\mathrm{emb}} = [\, z_1 C_1(D_{\boldsymbol{X}}, \boldsymbol{\mathcal{G}}), \ldots, z_d C_d(D_{\boldsymbol{X}}, \boldsymbol{\mathcal{G}}) \,] + P(D_{\boldsymbol{X}}, \boldsymbol{\mathcal{G}}).$$

 4: **Initialization:** Set initial latent state (which will be transformed into the estimated causal variable) using the dataset embedding:

$$n_0 = \mu'(D_{\boldsymbol{X}}, \boldsymbol{\mathcal{G}}) W_{n_0}.$$

 5: **Fixed-Point Iterations:** For $\ell = 0, \ldots, L-1$, apply DAG-Attention and adaptive normalization:

$$\boldsymbol{n}_{\ell+1} = h(\mathrm{DAM}(\boldsymbol{n}_\ell, \boldsymbol{z}_{\mathrm{emb}})\, \boldsymbol{z}_{\mathrm{emb}} + \boldsymbol{n}_\ell)\,,$$

where $h(\cdot)$ includes adaptive LayerNorm whose scale and shift depend on $\mu'(D_{\boldsymbol{X}}, \boldsymbol{\mathcal{G}})$.
 6: **Output Projection:** After $L$ iterations, project the final state:

$$\mathcal{T}(\boldsymbol{z}, D_{\boldsymbol{X}}, \boldsymbol{\mathcal{G}}) = \boldsymbol{n}_L W_{\mathrm{out}} \in \mathbb{R}^d.$$

 7: **Training Objective:** Minimize reconstruction loss over random samples $z \sim P_X$:

$$\mathcal{L}_T = \mathbb{E}_{S \sim P_S,\, \boldsymbol{z} \sim P_X}\left[\| \mathcal{T}(\boldsymbol{z}, D_{\boldsymbol{X}}, \boldsymbol{\mathcal{G}}) - \boldsymbol{F}(\boldsymbol{z}) \|_2^2\right].$$

---

# B   Details on Experiment Setup

## B.1   AVICI Benchmark

We use the synthetic data generation procedure proposed by Lorch et al. (2022) to generate SCMs in our empirical study. It provides access to a wide variety of SCMs, hence making it an excellent setting for amortized training.

- **Graphs:** We have the option to sample graphs as per the following schemes: Erods-Renyi (Erdos & Renyi, 1959), scale-free models (Barabási & Albert, 1999), Watts-Strogatz (Watts & Strogatz, 1998), and stochastic block models (Holland et al., 1983).

- **Noise Variables:** To sample noise variables, we can choose from either the gaussian or laplace distribution where variances are sampled randomly.

- **Functional Mechanisms:** We can control the complexity of causal relationships: either we set them to be linear (LIN) functions randomly sampled, or use random fourier features (RFF) for generating random non-linear causal relationships.

We construct two distribution of SCMs $\mathbb{P}_{IN}$, and $\mathbb{P}_{OUT}$, which vary based on the choice for sampling causal graphs, noise variables, and causal relationships. The classification aids in understanding the creation of train and test datasets.

- **In-Distribution ($\mathbb{P}_{IN}$):** We sample causal graphs using the Erods-Renyi and scale-free models schemes. Noise variables are sampled from the gaussian distribution, and we allow for both LIN and RFF causal relationships.

- **Out-of-Distribution ($\mathbb{P}_{OUT}$):** Causal graphs are drawn from Watts-Strogatz and stochastic block models schemes. Noise variables follow the laplace distribution, and both the LIN and RFF cases are used to sample functions. However, the parameters of these distributions are sampled from a different range as compared to $\mathbb{P}_{IN}$ to create a distribution shift.

We provide further details on the shift in the support of parameters for functional mechanisms below. For complete details please refer to Table 3, Appendix in Lorch et al. (2022).

- **Linear Functional Mechanism.**
    - *In-Distribution ($\mathbb{P}_{IN}$)*
        * Weights: $\sim U_{\pm}(1, 3)$, Bias $\sim U(-3, 3)$.
    - *Out-of-Distribution ($\mathbb{P}_{OUT}$)*
        * Weights: $\sim U_{\pm}(0.5, 2) \cup U_{\pm}(2, 4)$, Bias $\sim U(-3, 3)$.

- **RFF Functional Mechanism.**
    - *In-Distribution ($\mathbb{P}_{IN}$)*
        * Length Scale: $\sim U(7, 10)$, Output Scale: $\sim U(5, 8) \cup U(8, 12)$, Bias $\sim U_{\pm}(-3, 3)$.
    - *Out-of-Distribution ($\mathbb{P}_{OUT}$):*
        * Length Scale: $\sim U(10, 20)$, Output Scale: $\sim U(8, 12) \cup U(18, 22)$, Bias $\sim U_{\pm}(-3, 3)$.

**Test Datasets.**

- LIN **IN**: SCMs sampled from $\mathbb{P}_{IN}$ with linear causal mechanisms. We have 3 different options for sampling graphs in this case, and we randomly sample 3 different SCMs for each scenario, leading to a total of 9 instances.

- RFF **IN**: SCMs sampled from $\mathbb{P}_{IN}$ with non-linear causal mechanisms. We have 3 different options for sampling graphs in this case, and we randomly sample 3 different SCMs for each scenario, leading to a total of 9 instances.

- LIN **OUT**: SCMs sampled from $\mathbb{P}_{OUT}$ with linear causal mechanisms. We have 2 different options for sampling graphs in this case, and we randomly sample 3 different SCMs for each scenario, leading to a total of 6 instances.

- RFF **OUT**: SCMs sampled from $\mathbb{P}_{OUT}$ with non-linear causal mechanisms. We have 2 different options for sampling graphs in this case, and we randomly sample 3 different SCMs for each scenario, leading to a total of 6 instances.

## B.2   CSuite Benchmark

CSuite (Geffner et al., 2022) is a collection of synthetic structural causal models (SCMs) designed to evaluate causal discovery and effect estimation methods. The benchmark covers a diverse set of settings by varying graph structures, functional forms, and noise distributions, thereby testing models under both linear and nonlinear, Gaussian and non-Gaussian conditions, etc. We use the *lingauss*, *linexp*, *nonlingauss*, *nonlin-simpson*, *symprod-simpson*, *large-backdoor*, and *weak-arrow* tasks from their paper.

To conduct more OOD evaluations, we modify the noise distribution of the Large Backdoor and Weak Arrow datasets from the Csuite benchmark such that the noise variables are sampled from a guassian mixture model (GMM). We considered the following cases for the GMM noise distribution.

- Noise is sampled with equal probability from either $N(-2, 1)$ and $N(2, 1)$.

- Noise is sampled with equal probability from either $N(-2, 2)$ and $N(2, 2)$.

- Noise is sampled with equal probability from either $N(-2, 1)$ and $N(2, 2)$.

- Noise is sampled with equal probability from either $N(-5, 1)$ and $N(5, 1)$.

- Noise is sampled with equal probability from either $N(-5, 2)$ and $N(5, 2)$.

- Noise is sampled with equal probability from either $N(-5, 1)$ and $N(5, 2)$.

This leads to a total of 12 experimental setting with 6 different GMM noise distribution for both the Large Backdoor and Weak Arrow datasets from the CSuite benchmark.

### B.3 Model Architecture and Training Details

For both the dataset encoder and cond-FiP, we set the embedding dimension to $d_h = 256$ and the hidden dimension of MLP blocks to 512. Both of our transformer-based models contains 4 attention layers and each attention consists of 8 attention heads. The models were trained for a total of $10k$ epochs with the Adam optimizer (Paszke et al., 2017), where we used a learning rate of $1e-4$ and a weight decay of $5e-9$. Each epoch contains $\simeq 400$ randomly generated datasets from the distribution $\mathbb{P}_{\text{IN}}$. We also use the EMA implementation of (Karras et al., 2023) to train our models.

**Memory Requirements.** We trained Cond-FiP on a single L40 GPU with 48GB of memory, using an effective batch size of 8 with gradient accumulation. We outline the detailed memory computation as follows:

- Each batch consists of $n = 400$ samples with dimension $d = 20$ requiring less than 1 MiB of data in FP32 precision. Also, storing the model on the GPU requires under 100 MiB.

- Our transformer architecture has 4 attention layers, a 256-dimensional embedding space, and a 512-dimensional feedforward network. Using a standard (non-flash) attention implementation, a forward pass consumes approximately 30 GiB of GPU memory.

Compared to the baselines, Cond-FiP has similar memory requirements to DECI (Geffner et al., 2022) and FiP (Scetbon et al., 2024), as all three train neural networks of comparable size. The main exception is DoWhy (Blöbaum et al., 2022), which fits simpler models for each node, but this approach does not scale well as the graph size increases.

## C    Complete Results for Cond-FiP on AVICI Benchmark

| Method | Total Nodes | LIN **IN** | RFF **IN** | LIN **OUT** | RFF **OUT** |
|--------|------------|-----------|-----------|------------|------------|
| DoWhy | 10 | 0.03 (0.0) | 0.13 (0.02) | 0.04 (0.01) | 0.11 (0.01) |
| DECI | 10 | 0.09 (0.01) | 0.23 (0.03) | 0.12 (0.01) | 0.23 (0.03) |
| FiP | 10 | 0.04 (0.0) | 0.09 (0.01) | 0.06 (0.01) | 0.08 (0.01) |
| Cond-FiP | 10 | 0.06 (0.01) | 0.10 (0.01) | 0.07 (0.01) | 0.10 (0.01) |
| DoWhy | 20 | 0.03 (0.01) | 0.15 (0.02) | 0.03 (0.0) | 0.23 (0.01) |
| DECI | 20 | 0.10 (0.02) | 0.21 (0.03) | 0.08 (0.02) | 0.23 (0.02) |
| FiP | 20 | 0.04 (0.0) | 0.12 (0.02) | 0.05 (0.0) | 0.15 (0.02) |
| Cond-FiP | 20 | 0.06 (0.01) | 0.09 (0.01) | 0.07 (0.0) | 0.12 (0.0) |
| DoWhy | 50 | 0.03 (0.0) | 0.18 (0.03) | 0.03 (0.0) | 0.29 (0.03) |
| DECI | 50 | 0.09 (0.01) | 0.24 (0.02) | 0.07 (0.01) | 0.29 (0.02) |
| FiP | 50 | 0.04 (0.0) | 0.14 (0.03) | 0.04 (0.0) | 0.23 (0.04) |
| Cond-FiP | 50 | 0.06 (0.01) | 0.10 (0.01) | 0.07 (0.01) | 0.14 (0.01) |
| DoWhy | 100 | 0.03 (0.0) | 0.20 (0.03) | 0.03 (0.0) | 0.31 (0.02) |
| DECI | 100 | 0.08 (0.02) | 0.26 (0.03) | 0.07 (0.01) | 0.30 (0.02) |
| FiP | 100 | 0.04 (0.0) | 0.16 (0.03) | 0.04 (0.0) | 0.24 (0.02) |
| Cond-FiP | 100 | 0.05 (0.0) | 0.10 (0.01) | 0.07 (0.01) | 0.16 (0.01) |

Table 3: **Results for Noise Prediction.** We compare Cond-FiP against the baselines for the task of predicting noise variables from the input observations. Each cell reports the mean (standard error) RMSE over the multiple test datasets for each scenario. Shaded rows denote the case where the graph size is larger than the train graph sizes ($d = 20$) for Cond-FiP. *Results show that Cond-FiP generalizes to both in-distribution and OOD instances.*

| Method | Total Nodes | LIN **IN** | RFF **IN** | LIN **OUT** | RFF **OUT** |
|---|---|---|---|---|---|
| DoWhy | 10 | 0.05 (0.0) | 0.18 (0.03) | 0.06 (0.01) | 0.12 (0.02) |
| DECI | 10 | 0.15 (0.02) | 0.33 (0.04) | 0.16 (0.02) | 0.27 (0.03) |
| FiP | 10 | 0.07 (0.0) | 0.13 (0.02) | 0.08 (0.01) | 0.11 (0.02) |
| Cond-FiP | 10 | 0.06 (0.01) | 0.14 (0.02) | 0.05 (0.01) | 0.08 (0.01) |
| DoWhy | 20 | 0.06 (0.01) | 0.27 (0.05) | 0.05 (0.0) | 0.39 (0.04) |
| DECI | 20 | 0.16 (0.02) | 0.39 (0.05) | 0.13 (0.02) | 0.44 (0.04) |
| FiP | 20 | 0.08 (0.01) | 0.23 (0.05) | 0.08 (0.01) | 0.27 (0.04) |
| Cond-FiP | 20 | 0.05 (0.01) | 0.24 (0.06) | 0.07 (0.01) | 0.30 (0.03) |
| DoWhy | 50 | 0.08 (0.01) | 0.35 (0.09) | 0.06 (0.01) | 0.54 (0.06) |
| DECI | 50 | 0.15 (0.01) | 0.46 (0.06) | 0.13 (0.02) | 0.67 (0.06) |
| FiP | 50 | 0.09 (0.01) | 0.26 (0.05) | 0.08 (0.01) | 0.48 (0.06) |
| Cond-FiP | 50 | 0.08 (0.01) | 0.25 (0.05) | 0.07 (0.0) | 0.48 (0.07) |
| DoWhy | 100 | 0.06 (0.0) | 0.33 (0.07) | 0.06 (0.01) | 0.63 (0.07) |
| DECI | 100 | 0.14 (0.02) | 0.50 (0.09) | 0.14 (0.02) | 0.71 (0.08) |
| FiP | 100 | 0.08 (0.01) | 0.3 (0.06) | 0.09 (0.01) | 0.55 (0.08) |
| Cond-FiP | 100 | 0.07 (0.01) | 0.29 (0.07) | 0.09 (0.01) | 0.57 (0.07) |

Table 4: **Results for Sample Generation.** We compare Cond-FiP against the baselines for the task of generating samples from the input noise variables. Each cell reports the mean (standard error) RMSE over the multiple test datasets for each scenario. Shaded rows denote the case where the graph size is larger than the train graph sizes ($d = 20$) for Cond-FiP. *Results show that Cond-FiP generalizes to both in-distribution and OOD instances.*

| Method | Total Nodes | LIN **IN** | RFF **IN** | LIN **OUT** | RFF **OUT** |
|---|---|---|---|---|---|
| DoWhy | 10 | 0.08 (0.03) | 0.19 (0.04) | 0.05 (0.01) | 0.12 (0.02) |
| DECI | 10 | 0.17 (0.02) | 0.34 (0.04) | 0.13 (0.02) | 0.25 (0.03) |
| FiP | 10 | 0.08 (0.01) | 0.15 (0.02) | 0.07 (0.01) | 0.09 (0.01) |
| Cond-FiP | 10 | 0.10 (0.03) | 0.21 (0.03) | 0.07 (0.01) | 0.11 (0.01) |
| DoWhy | 20 | 0.06 (0.01) | 0.27 (0.06) | 0.05 (0.0) | 0.36 (0.03) |
| DECI | 20 | 0.16 (0.02) | 0.38 (0.05) | 0.15 (0.04) | 0.42 (0.03) |
| FiP | 20 | 0.09 (0.01) | 0.23 (0.05) | 0.12 (0.04) | 0.25 (0.03) |
| Cond-FiP | 20 | 0.09 (0.01) | 0.24 (0.05) | 0.14 (0.03) | 0.31 (0.03) |
| DoWhy | 50 | 0.08 (0.01) | 0.29 (0.05) | 0.06 (0.01) | 0.53 (0.06) |
| DECI | 50 | 0.17 (0.02) | 0.44 (0.06) | 0.13 (0.02) | 0.64 (0.06) |
| FiP | 50 | 0.11 (0.02) | 0.25 (0.05) | 0.09 (0.01) | 0.46 (0.06) |
| Cond-FiP | 50 | 0.13 (0.02) | 0.27 (0.04) | 0.12 (0.02) | 0.48 (0.07) |
| DoWhy | 100 | 0.05 (0.0) | 0.33 (0.07) | 0.06 (0.01) | 0.60 (0.07) |
| DECI | 100 | 0.14 (0.02) | 0.49 (0.08) | 0.15 (0.02) | 0.70 (0.08) |
| FiP | 100 | 0.08 (0.01) | 0.29 (0.07) | 0.10 (0.01) | 0.54 (0.08) |
| Cond-FiP | 100 | 0.10 (0.01) | 0.30 (0.06) | 0.14 (0.02) | 0.56 (0.07) |

Table 5: **Results for Interventional Generation.** We compare Cond-FiP against the baselines for the task of generating interventional data from the input noise variables. Each cell reports the mean (standard error) RMSE over the multiple test datasets for each scenario. Shaded rows denote the case where the graph size is larger than the train graph sizes ($d = 20$) for Cond-FiP. *Results show that Cond-FiP generalizes to both in-distribution and OOD instances.*

# D   Experiments on Sensitivity to Distribution Shifts on AVICI benchmark

In Appendix C (Table 3, Table 4, Table 5), we tested OOD genrealization with datasets sampled from SCM following a different distribution (LIN **OUT**, RFF **OUT**) than the datasets used for training Cond-FiP (LIN **IN**, RFF **IN**). We now analyze how sensitive is Cond-FiP to distribution shifts by comparing its performance across scenarios as the severity of the distribution shift is increased.

To illustrate how we control the magnitude of distribution shift, we discuss the difference in the distribution of causal mechanisms across $\mathbb{P}_{\text{IN}}$ and $\mathbb{P}_{\text{OUT}}$. The distribution shift arises because the support of the parameters of causal mechanisms changes from $\mathbb{P}_{\text{IN}}$ to $\mathbb{P}_{\text{OUT}}$. For example, for linear causal mechanism case, the weights in $\mathbb{P}_{\text{IN}}$ are sampled uniformly from $(-3, -1) \cup (1, 3)$; while in $\mathbb{P}_{\text{OUT}}$ they are sampled from uniformly from $(0.5, 4)$. We now change the support set of the parameters in $\mathbb{P}_{\text{OUT}}$ to $(0.5\alpha, 4\alpha)$, so that by increasing $\alpha$ we make the distribution shift more severe. We follow this procedure for the support set of all the parameters associated with functional mechanisms and generate distributions $(\mathbb{P}_{\text{OUT}}(\alpha))$ with varying shift w.r.t $\mathbb{P}_{\text{IN}}$ by changing $\alpha$. Note that $\alpha = 1$ corresponds to the same $\mathbb{P}_{\text{OUT}}$ as the one used for sampling datasets in our main results.

We conduct two experiments for evaluating the robustness of Cond-FiP to distribution shifts, described ahead.

- **Controlling Shift in Causal Mechanisms.** We start with the parameter configuration of $\mathbb{P}_{\text{OUT}}$ from the setup in main results; and then control the magnitude of shift by changing the support set of parameters of causal mechanisms.

- **Controlling Shift in Noise Variables.** We start with the parameter configuration of $\mathbb{P}_{\text{OUT}}$ from the setup in main results; and then control the magnitude of shift by changing the support set of parameters of noise distribution.

Tables 6, 7, and 8 provide results for the case of controlling shift via causal mechanisms, for the task of noise prediction, sample generation, and interventional generation respectively. We find that the performance of Cond-FiP does not change much as we increase $\alpha$, indicating that Cond-FiP is robust to the varying levels of distribution shits in causal mechanisms.

However, for the case of controlling shift via noise variables (Table 9, 10, and 11) we find that Cond-FiP is quite sensitive to the varying levels of distribution shift in noise variables. The performance of Cond-FiP degrades with increasing magnitude of the shift ($\alpha$) for all the tasks.

| Total Nodes | Shift Level ($\alpha$) | LIN **OUT** | RFF **OUT** |
|:---:|:---:|:---:|:---:|
| 10 | 1 | 0.07 (0.01) | 0.10 (0.01) |
| 10 | 2 | 0.06 (0.01) | 0.10 (0.01) |
| 10 | 5 | 0.05 (0.01) | 0.10 (0.01) |
| 10 | 10 | 0.05 (0.01) | 0.10 (0.01) |
| 20 | 1 | 0.07 (0.0) | 0.12 (0.0) |
| 20 | 2 | 0.06 (0.0) | 0.13 (0.01) |
| 20 | 5 | 0.05 (0.0) | 0.11 (0.01) |
| 20 | 10 | 0.05 (0.0) | 0.10 (0.01) |
| 50 | 1 | 0.07 (0.01) | 0.14 (0.01) |
| 50 | 2 | 0.05 (0.01) | 0.17 (0.01) |
| 50 | 5 | 0.05 (0.01) | 0.14 (0.01) |
| 50 | 10 | 0.04 (0.0) | 0.14 (0.01) |
| 100 | 1 | 0.07 (0.01) | 0.16 (0.01) |
| 100 | 2 | 0.05 (0.01) | 0.18 (0.0) |
| 100 | 5 | 0.05 (0.0) | 0.17 (0.01) |
| 100 | 10 | 0.05 (0.0) | 0.16 (0.01) |

Table 6: **Results for Noise Prediction under Distribution Shifts in Causal Mechanisms.** We evaluate the robustness of Cond-FiP to distribution shifts in the parametrization of causal mechanisms. We vary the distribution shift controlled by $\alpha$, where $\alpha = 1$ corresponds to the results in Table 3. Each cell reports the mean (standard error) RMSE over the multiple test datasets for each scenario. *We find that Cond-FiP is robust to varying levels of distribution shift in causal mechanisms.*

| Total Nodes | Shift Level ($\alpha$) | LIN **OUT** | RFF **OUT** |
|:---:|:---:|:---:|:---:|
| 10 | 1 | 0.05 (0.01) | 0.08 (0.01) |
| 10 | 2 | 0.05 (0.0) | 0.07 (0.01) |
| 10 | 5 | 0.05 (0.0) | 0.07 (0.01) |
| 10 | 10 | 0.06 (0.0) | 0.06 (0.01) |
| 20 | 1 | 0.07 (0.01) | 0.30 (0.03) |
| 20 | 2 | 0.06 (0.01) | 0.34 (0.05) |
| 20 | 5 | 0.06 (0.01) | 0.35 (0.05) |
| 20 | 10 | 0.06 (0.01) | 0.29 (0.07) |
| 50 | 1 | 0.07 (0.0) | 0.48 (0.07) |
| 50 | 2 | 0.07 (0.0) | 0.47 (0.07) |
| 50 | 5 | 0.07 (0.01) | 0.38 (0.06) |
| 50 | 10 | 0.07 (0.01) | 0.32 (0.06) |
| 100 | 1 | 0.09 (0.01) | 0.57 (0.07) |
| 100 | 2 | 0.09 (0.01) | 0.60 (0.05) |
| 100 | 5 | 0.09 (0.01) | 0.58 (0.05) |
| 100 | 10 | 0.12 (0.02) | 0.56 (0.06) |

Table 7: **Results for Sample Generation under Distribution Shifts in Causal Mechanisms.** We evaluate the robustness of Cond-FiP to distribution shifts in the parametrization of causal mechanisms. We vary the distribution shift controlled by $\alpha$, where $\alpha = 1$ corresponds to the results in Table 4. Each cell reports the mean (standard error) RMSE over the multiple test datasets for each scenario. *We find that Cond-FiP is robust to varying levels of distribution shift in causal mechanisms.*

| Total Nodes | Shift Level ($\alpha$) | LIN **OUT** | RFF **OUT** |
|:---:|:---:|:---:|:---:|
| 10 | 1 | 0.07 (0.01) | 0.11 (0.01) |
| 10 | 2 | 0.07 (0.01) | 0.11 (0.01) |
| 10 | 5 | 0.07 (0.01) | 0.10 (0.01) |
| 10 | 10 | 0.06 (0.01) | 0.10 (0.01) |
| 20 | 1 | 0.14 (0.03) | 0.31 (0.03) |
| 20 | 2 | 0.10 (0.02) | 0.33 (0.04) |
| 20 | 5 | 0.17 (0.1) | 0.34 (0.04) |
| 20 | 10 | 0.10 (0.03) | 0.28 (0.05) |
| 50 | 1 | 0.12 (0.02) | 0.48 (0.07) |
| 50 | 2 | 0.12 (0.03) | 0.47 (0.07) |
| 50 | 5 | 0.11 (0.01) | 0.39 (0.06) |
| 50 | 10 | 0.11 (0.02) | 0.32 (0.06) |
| 100 | 1 | 0.14 (0.02) | 0.58 (0.07) |
| 100 | 2 | 0.13 (0.02) | 0.60 (0.06) |
| 100 | 5 | 0.14 (0.03) | 0.58 (0.05) |
| 100 | 10 | 0.18 (0.04) | 0.55 (0.06) |

Table 8: **Results for Interventional Generation under Distribution Shifts in Causal Mechanisms.** We evaluate the robustness of Cond-FiP to distribution shifts in the parametrization of causal mechanisms. We vary the distribution shift controlled by $\alpha$, where $\alpha = 1$ corresponds to the results in Table 5. Each cell reports the mean (standard error) RMSE over the multiple test datasets for each scenario. *We find that Cond-FiP is robust to varying levels of distribution shift in causal mechanisms.*

| Total Nodes | Shift Level ($\alpha$) | LIN **OUT** | RFF **OUT** |
|:---:|:---:|:---:|:---:|
| 10 | 1 | 0.07 (0.01) | 0.10 (0.01) |
| 10 | 2 | 0.07 (0.01) | 0.11 (0.01) |
| 10 | 5 | 0.07 (0.01) | 0.18 (0.02) |
| 10 | 10 | 0.08 (0.01) | 0.26 (0.04) |
| 20 | 1 | 0.07 (0.0) | 0.12 (0.0) |
| 20 | 2 | 0.07 (0.0) | 0.16 (0.01) |
| 20 | 5 | 0.07 (0.0) | 0.30 (0.01) |
| 20 | 10 | 0.07 (0.0) | 0.41 (0.02) |
| 50 | 1 | 0.07 (0.01) | 0.14 (0.01) |
| 50 | 2 | 0.07 (0.01) | 0.19 (0.01) |
| 50 | 5 | 0.07 (0.01) | 0.33 (0.02) |
| 50 | 10 | 0.07 (0.01) | 0.44 (0.02) |
| 100 | 1 | 0.07 (0.01) | 0.16 (0.01) |
| 100 | 2 | 0.07 (0.01) | 0.22 (0.0) |
| 100 | 5 | 0.07 (0.01) | 0.35 (0.01) |
| 100 | 10 | 0.07 (0.01) | 0.44 (0.01) |

Table 9: **Results for Noise Prediction under Distribution Shifts in Noise Variables.** We evaluate the robustness of Cond-FiP to distribution shifts in the parametrization of noise distribution. We vary the distribution shift controlled by $\alpha$, where $\alpha = 1$ corresponds to the results in Table 3. Each cell reports the mean (standard error) RMSE over the multiple test datasets for each scenario. *We find that Cond-FiP is sensitive to varying levels of distribution shift in noise variables, its performance decreases with increasing magnitude of the shift.*

| Total Nodes | Shift Level ($\alpha$) | LIN **OUT** | RFF **OUT** |
|:---:|:---:|:---:|:---:|
| 10 | 1 | 0.05 (0.01) | 0.08 (0.01) |
| 10 | 2 | 0.05 (0.0) | 0.13 (0.03) |
| 10 | 5 | 0.05 (0.01) | 0.28 (0.06) |
| 10 | 10 | 0.05 (0.01) | 0.36 (0.08) |
| 20 | 1 | 0.07 (0.01) | 0.30 (0.03) |
| 20 | 2 | 0.07 (0.01) | 0.45 (0.04) |
| 20 | 5 | 0.07 (0.01) | 0.59 (0.03) |
| 20 | 10 | 0.07 (0.01) | 0.58 (0.02) |
| 50 | 1 | 0.07 (0.0) | 0.48 (0.07) |
| 50 | 2 | 0.07 (0.0) | 0.59 (0.06) |
| 50 | 5 | 0.07 (0.0) | 0.64 (0.03) |
| 50 | 10 | 0.07 (0.0) | 0.58 (0.02) |
| 100 | 1 | 0.09 (0.01) | 0.57 (0.07) |
| 100 | 2 | 0.09 (0.01) | 0.63 (0.05) |
| 100 | 5 | 0.09 (0.01) | 0.65 (0.03) |
| 100 | 10 | 0.09 (0.01) | 0.59 (0.02) |

Table 10: **Results for Sample Generation under Distribution Shifts in Noise Variables.** We evaluate the robustness of Cond-FiP to distribution shifts in the parametrization of noise distribution. We vary the distribution shift controlled by $\alpha$, where $\alpha = 1$ corresponds to the results in Table 4. Each cell reports the mean (standard error) RMSE over the multiple test datasets for each scenario. *We find that Cond-FiP is sensitive to varying levels of distribution shift in noise variables, its performance decreases with increasing magnitude of the shift.*

| Total Nodes | Shift Level ($\alpha$) | LIN **OUT** | RFF **OUT** |
|:---:|:---:|:---:|:---:|
| 10 | 1 | 0.07 (0.01) | 0.11 (0.01) |
| 10 | 2 | 0.07 (0.01) | 0.14 (0.02) |
| 10 | 5 | 0.07 (0.01) | 0.25 (0.05) |
| 10 | 10 | 0.07 (0.01) | 0.32 (0.06) |
| 20 | 1 | 0.14 (0.03) | 0.31 (0.03) |
| 20 | 2 | 0.14 (0.03) | 0.42 (0.03) |
| 20 | 5 | 0.14 (0.03) | 0.57 (0.03) |
| 20 | 10 | 0.14 (0.03) | 0.56 (0.02) |
| 50 | 1 | 0.12 (0.02) | 0.48 (0.07) |
| 50 | 2 | 0.12 (0.01) | 0.58 (0.06) |
| 50 | 5 | 0.12 (0.01) | 0.65 (0.04) |
| 50 | 10 | 0.12 (0.01) | 0.59 (0.02) |
| 100 | 1 | 0.14 (0.02) | 0.58 (0.07) |
| 100 | 2 | 0.14 (0.02) | 0.65 (0.06) |
| 100 | 5 | 0.14 (0.02) | 0.67 (0.04) |
| 100 | 10 | 0.14 (0.02) | 0.60 (0.03) |

Table 11: **Results for Interventional Generation under Distribution Shifts in Noise Variables.** We evaluate the robustness of Cond-FiP to distribution shifts in the parametrization of noise distribution. We vary the distribution shift controlled by $\alpha$, where $\alpha = 1$ corresponds to the results in Table 5. Each cell reports the mean (standard error) RMSE over the multiple test datasets for each scenario. *We find that Cond-FiP is sensitive to varying levels of distribution shift in noise variables, its performance decreases with increasing magnitude of the shift.*

# E   Experiment on Generalization in Scarce Data Regime on AVICI benchmark

## E.1   Experiments with $n_{\mathcal{D}_{\text{test}}} = 100$

In this section we benchmark Cond-FiP against the baselines for the scenario when test datasets in the input context have smaller sample size ($n_{\mathcal{D}_{\text{test}}} = 100$) as compared to the train datasets ($n_{\mathcal{D}_{\text{test}}} = 400$) in Appendix C.

We report the results for the task of noise prediction, sample generation, and interventional generation in Table 12, Table 13, and Table 14 respectively. We find that Cond-FiP exhibits superior generalization as compared to baselines. For example, in the case of RFF **IN**, Cond-FiP is even better than FiP for all the tasks! This can be attributed to the advantage of amortized inference; as the sample size in test dataset decreases, the generalization of baselines would be affected a lot since they require training from scratch on these datasets. However, amortized inference methods would be impacted less as they do not have to trained from scratch, and the inductive bias learned by them can help them generalize even with smaller input context.

| Method | Total Nodes | LIN **IN** | RFF **IN** | LIN **OUT** | RFF **OUT** |
|---|---|---|---|---|---|
| DoWhy | 10 | 0.06 (0.01) | 0.22 (0.03) | 0.09 (0.01) | 0.16 (0.03) |
| DECI | 10 | 0.15 (0.01) | 0.3 (0.02) | 0.22 (0.01) | 0.3 (0.03) |
| FiP | 10 | 0.07 (0.01) | 0.18 (0.01) | 0.12 (0.01) | 0.11 (0.01) |
| Cond-FiP | 10 | 0.07 (0.01) | 0.14 (0.01) | 0.09 (0.01) | 0.14 (0.01) |
| DoWhy | 20 | 0.06 (0.01) | 0.27 (0.05) | 0.07 (0.01) | 0.37 (0.01) |
| DECI | 20 | 0.15 (0.02) | 0.33 (0.02) | 0.17 (0.02) | 0.35 (0.03) |
| FiP | 20 | 0.09 (0.01) | 0.21 (0.03) | 0.1 (0.01) | 0.27 (0.03) |
| Cond-FiP | 20 | 0.08 (0.01) | 0.12 (0.01) | 0.1 (0.01) | 0.15 (0.01) |
| DoWhy | 50 | 0.06 (0.01) | 0.29 (0.04) | 0.05 (0.01) | 0.47 (0.04) |
| DECI | 50 | 0.14 (0.01) | 0.33 (0.02) | 0.14 (0.02) | 0.4 (0.03) |
| FiP | 50 | 0.08 (0.01) | 0.23 (0.03) | 0.08 (0.01) | 0.37 (0.04) |
| Cond-FiP | 50 | 0.08 (0.0) | 0.12 (0.01) | 0.08 (0.01) | 0.15 (0.01) |
| DoWhy | 100 | 0.06 (0.01) | 0.31 (0.04) | 0.06 (0.01) | 0.5 (0.03) |
| DECI | 100 | 0.13 (0.01) | 0.36 (0.03) | 0.12 (0.02) | 0.44 (0.02) |
| FiP | 100 | 0.08 (0.01) | 0.25 (0.04) | 0.1 (0.01) | 0.39 (0.03) |
| Cond-FiP | 100 | 0.07 (0.0) | 0.13 (0.01) | 0.08 (0.01) | 0.17 (0.01) |

Table 12: **Results for Noise Prediction with Smaller Sample Size ($n_{\mathcal{D}_{\text{test}}} = 100$).** We compare Cond-FiP against the baselines for the task of predicting noise variable from input observations. Each test dataset contains 100 samples, as opposed to 400 samples in Table 3. Each cell reports the mean (standard error) RMSE over the multiple test datasets for each scenario. Shaded rows deonte the case where the graph size is larger than the train graph sizes ($d = 20$) for Cond-FiP. *Results show that Cond-FiP generalizes much better than the baselines in this low-data regime.*

| Method | Total Nodes | LIN **IN** | RFF **IN** | LIN **OUT** | RFF **OUT** |
|---|---|---|---|---|---|
| DoWhy | 10 | 0.1 (0.01) | 0.3 (0.06) | 0.12 (0.02) | 0.19 (0.03) |
| DECI | 10 | 0.23 (0.01) | 0.45 (0.04) | 0.31 (0.02) | 0.38 (0.04) |
| FiP | 10 | 0.13 (0.01) | 0.29 (0.04) | 0.18 (0.02) | 0.15 (0.03) |
| Cond-FiP | 10 | 0.09 (0.01) | 0.2 (0.03) | 0.09 (0.02) | 0.14 (0.02) |
| DoWhy | 20 | 0.11 (0.01) | 0.47 (0.15) | 0.11 (0.02) | 0.5 (0.03) |
| DECI | 20 | 0.26 (0.02) | 0.53 (0.05) | 0.26 (0.03) | 0.57 (0.04) |
| FiP | 20 | 0.17 (0.02) | 0.34 (0.06) | 0.17 (0.02) | 0.39 (0.03) |
| Cond-FiP | 20 | 0.08 (0.0) | 0.31 (0.06) | 0.13 (0.01) | 0.37 (0.02) |
| DoWhy | 50 | 0.11 (0.01) | 0.42 (0.08) | 0.09 (0.01) | 0.66 (0.06) |
| DECI | 50 | 0.23 (0.02) | 0.59 (0.08) | 0.27 (0.04) | 0.73 (0.06) |
| FiP | 50 | 0.13 (0.01) | 0.38 (0.07) | 0.14 (0.01) | 0.58 (0.06) |
| Cond-FiP | 50 | 0.1 (0.01) | 0.32 (0.05) | 0.12 (0.01) | 0.54 (0.05) |
| DoWhy | 100 | 0.11 (0.01) | 0.44 (0.08) | 0.11 (0.01) | 0.74 (0.05) |
| DECI | 100 | 0.25 (0.02) | 0.62 (0.08) | 0.25 (0.01) | 0.78 (0.07) |
| FiP | 100 | 0.15 (0.01) | 0.4 (0.07) | 0.19 (0.02) | 0.67 (0.07) |
| Cond-FiP | 100 | 0.11 (0.01) | 0.35 (0.07) | 0.14 (0.02) | 0.63 (0.07) |

Table 13: **Results for Sample Generation with Smaller Sample Size ($n_{\mathcal{D}_{\text{test}}} = 100$).** We compare Cond-FiP against the baselines for the task of generating samples from the input noise variable. Each test dataset contains 100 samples, as opposed to 400 samples in Table 4. Each cell reports the mean (standard error) RMSE over the multiple test datasets for each scenario. Shaded rows deonte the case where the graph size is larger than the train graph sizes ($d = 20$) for Cond-FiP. *Results show that Cond-FiP generalizes much better than the baselines in this low-data regime.*

| Method | Total Nodes | LIN **IN** | RFF **IN** | LIN **OUT** | RFF **OUT** |
|---|---|---|---|---|---|
| DoWhy | 10 | 0.09 (0.01) | 0.34 (0.08) | 0.11 (0.01) | 0.2 (0.04) |
| DECI | 10 | 0.24 (0.02) | 0.43 (0.04) | 0.26 (0.03) | 0.35 (0.04) |
| FiP | 10 | 0.13 (0.01) | 0.29 (0.04) | 0.14 (0.02) | 0.14 (0.03) |
| Cond-FiP | 10 | 0.09 (0.02) | 0.21 (0.03) | 0.09 (0.01) | 0.12 (0.02) |
| DoWhy | 20 | 0.1 (0.01) | 0.37 (0.08) | 0.11 (0.02) | 0.49 (0.04) |
| DECI | 20 | 0.25 (0.03) | 0.5 (0.05) | 0.28 (0.03) | 0.54 (0.04) |
| FiP | 20 | 0.16 (0.01) | 0.33 (0.06) | 0.2 (0.03) | 0.38 (0.03) |
| Cond-FiP | 20 | 0.1 (0.01) | 0.27 (0.05) | 0.15 (0.02) | 0.29 (0.03) |
| DoWhy | 50 | 0.12 (0.02) | 0.49 (0.14) | 0.09 (0.01) | 0.64 (0.07) |
| DECI | 50 | 0.26 (0.03) | 0.56 (0.07) | 0.26 (0.03) | 0.72 (0.06) |
| FiP | 50 | 0.16 (0.02) | 0.36 (0.06) | 0.15 (0.01) | 0.57 (0.06) |
| Cond-FiP | 50 | 0.13 (0.02) | 0.29 (0.04) | 0.12 (0.01) | 0.49 (0.07) |
| DoWhy | 100 | 0.11 (0.01) | 0.46 (0.07) | 0.11 (0.01) | 1.16 (0.38) |
| DECI | 100 | 0.24 (0.02) | 0.62 (0.08) | 0.26 (0.01) | 0.78 (0.07) |
| FiP | 100 | 0.16 (0.02) | 0.39 (0.07) | 0.2 (0.02) | 0.66 (0.07) |
| Cond-FiP | 100 | 0.12 (0.02) | 0.32 (0.07) | 0.13 (0.01) | 0.58 (0.07) |

Table 14: **Results for Interventional Generation with Smaller Sample Size ($n_{\mathcal{D}_{\text{test}}} = 100$).** We compare Cond-FiP against the baselines for the task of generating interventional data from the input noise variable. Each test dataset contains 100 samples, as opposed to 400 samples in Table 5. Each cell reports the mean (standard error) RMSE over the multiple test datasets for each scenario. Shaded rows deonte the case where the graph size is larger than the train graph sizes ($d = 20$) for Cond-FiP. *Results show that Cond-FiP generalizes much better than the baselines in this low-data regime.*

### E.2 Experiments with $n_{\mathcal{D}_{\text{test}}} = 50$

We conduct more experiments for the smaller sample size scenarios, where decrease the sample size even further to $n_{\mathcal{D}_{\text{test}}} = 50$ samples. We report the results for the task of noise prediction, sample generation, and interventional generation in Table 15, Table 16, and Table 17 respectively. We find that baselines perform much worse than Cond-FiP for the all different SCM distributions, highlighting the efficacy of Cond-FiP for inferring causal mechanisms when the input context has smaller sample size. Note that there were issues with training DoWhy for such a small dataset, hence we do not consider them for this scenario.

| Method | Total Nodes | LIN **IN** | RFF **IN** | LIN **OUT** | RFF **OUT** |
|---|---|---|---|---|---|
| DECI | 10 | 0.19 (0.02) | 0.41 (0.03) | 0.2 (0.02) | 0.42 (0.04) |
| FiP | 10 | 0.13 (0.03) | 0.27 (0.03) | 0.15 (0.02) | 0.21 (0.03) |
| Cond-FiP | 10 | 0.09 (0.01) | 0.17 (0.01) | 0.11 (0.01) | 0.16 (0.01) |
| DECI | 20 | 0.2 (0.01) | 0.42 (0.03) | 0.25 (0.04) | 0.45 (0.05) |
| FiP | 20 | 0.12 (0.01) | 0.33 (0.04) | 0.15 (0.02) | 0.35 (0.04) |
| Cond-FiP | 20 | 0.1 (0.01) | 0.16 (0.01) | 0.11 (0.01) | 0.17 (0.01) |
| DECI | 50 | 0.2 (0.02) | 0.43 (0.02) | 0.2 (0.03) | 0.5 (0.05) |
| FiP | 50 | 0.13 (0.01) | 0.32 (0.03) | 0.13 (0.01) | 0.49 (0.05) |
| Cond-FiP | 50 | 0.1 (0.01) | 0.16 (0.0) | 0.1 (0.01) | 0.17 (0.01) |
| DECI | 100 | 0.19 (0.02) | 0.43 (0.03) | 0.21 (0.01) | 0.53 (0.02) |
| FiP | 100 | 0.11 (0.01) | 0.32 (0.04) | 0.13 (0.01) | 0.48 (0.02) |
| Cond-FiP | 100 | 0.09 (0.01) | 0.16 (0.01) | 0.09 (0.01) | 0.18 (0.01) |

Table 15: **Results for Noise Prediction with Smaller Sample Size ($n_{\mathcal{D}_{\text{test}}} = 50$).** We compare Cond-FiP against the baselines for the task of predicting noise variable from input observations. Each test dataset contains 50 samples, as opposed to 400 samples in Table 3. Each cell reports the mean (standard error) RMSE over the multiple test datasets for each scenario. Shaded rows denote the case where the graph size is larger than the train graph sizes ($d = 20$) for Cond-FiP. *Results show that Cond-FiP generalizes much better than the baselines in this low-data regime.*

| Method | Total Nodes | LIN **IN** | RFF **IN** | LIN **OUT** | RFF **OUT** |
|---|---|---|---|---|---|
| DECI | 10 | 0.31 (0.02) | 0.58 (0.05) | 0.27 (0.04) | 0.49 (0.07) |
| FiP | 10 | 0.2 (0.03) | 0.4 (0.05) | 0.21 (0.03) | 0.25 (0.04) |
| Cond-FiP | 10 | 0.12 (0.02) | 0.28 (0.03) | 0.12 (0.01) | 0.18 (0.03) |
| DECI | 20 | 0.34 (0.02) | 0.66 (0.08) | 0.39 (0.07) | 0.68 (0.05) |
| FiP | 20 | 0.2 (0.01) | 0.51 (0.08) | 0.25 (0.04) | 0.51 (0.02) |
| Cond-FiP | 20 | 0.13 (0.01) | 0.4 (0.06) | 0.19 (0.02) | 0.43 (0.02) |
| DECI | 50 | 0.32 (0.02) | 0.66 (0.06) | 0.36 (0.02) | 0.8 (0.06) |
| FiP | 50 | 0.2 (0.01) | 0.48 (0.07) | 0.22 (0.02) | 0.69 (0.06) |
| Cond-FiP | 50 | 0.15 (0.02) | 0.4 (0.05) | 0.16 (0.01) | 0.59 (0.06) |
| DECI | 100 | 0.36 (0.04) | 0.68 (0.08) | 0.39 (0.03) | 0.84 (0.06) |
| FiP | 100 | 0.2 (0.02) | 0.49 (0.09) | 0.28 (0.03) | 0.73 (0.07) |
| Cond-FiP | 100 | 0.16 (0.01) | 0.42 (0.07) | 0.22 (0.01) | 0.65 (0.06) |

Table 16: **Results for Sample Generation with Smaller Sample Size ($n_{\mathcal{D}_{\text{test}}} = 50$).** We compare Cond-FiP against the baselines for the task of generating samples from the input noise variable. Each test dataset contains 50 samples, as opposed to 400 samples in Table 4. Each cell reports the mean (standard error) RMSE over the multiple test datasets for each scenario. Shaded rows denote the case where the graph size is larger than the train graph sizes ($d = 20$) for Cond-FiP. *Results show that Cond-FiP generalizes much better than the baselines in this low-data regime.*

| Method | Total Nodes | LIN **IN** | RFF **IN** | LIN **OUT** | RFF **OUT** |
|---|---|---|---|---|---|
| DECI | 10 | 0.3 (0.03) | 0.53 (0.05) | 0.26 (0.04) | 0.42 (0.05) |
| FiP | 10 | 0.21 (0.04) | 0.35 (0.04) | 0.2 (0.03) | 0.22 (0.03) |
| Cond-FiP | 10 | 0.12 (0.01) | 0.19 (0.03) | 0.07 (0.01) | 0.14 (0.02) |
| DECI | 20 | 0.33 (0.02) | 0.6 (0.06) | 0.43 (0.07) | 0.63 (0.04) |
| FiP | 20 | 0.21 (0.02) | 0.46 (0.07) | 0.29 (0.04) | 0.49 (0.02) |
| Cond-FiP | 20 | 0.11 (0.01) | 0.29 (0.06) | 0.15 (0.02) | 0.32 (0.03) |
| DECI | 50 | 0.34 (0.02) | 0.66 (0.07) | 0.34 (0.02) | 0.78 (0.06) |
| FiP | 50 | 0.21 (0.02) | 0.46 (0.07) | 0.23 (0.02) | 0.68 (0.06) |
| Cond-FiP | 50 | 0.13 (0.02) | 0.31 (0.05) | 0.12 (0.02) | 0.51 (0.07) |
| DECI | 100 | 0.37 (0.04) | 0.67 (0.08) | 0.4 (0.04) | 0.84 (0.06) |
| FiP | 100 | 0.21 (0.02) | 0.49 (0.08) | 0.28 (0.03) | 0.73 (0.07) |
| Cond-FiP | 100 | 0.12 (0.01) | 0.33 (0.07) | 0.14 (0.01) | 0.58 (0.07) |

Table 17: **Results for Interventional Generation with Smaller Sample Size ($n_{\mathcal{D}_{\text{test}}} = 50$).** We compare Cond-FiP against the baselines for the task of generating interventional data from the input noise variable. Each test dataset contains 50 samples, as opposed to 400 samples in Table 5. Each cell reports the mean (standard error) RMSE over the multiple test datasets for each scenario. Shaded rows deonte the case where the graph size is larger than the train graph sizes ($d = 20$) for Cond-FiP. *Results show that Cond-FiP generalizes much better than the baselines in this low-data regime.*

# F   Experiments without True Causal Graph on AVICI Benchmark

Results in Appendix C (Table 3, Table 4, Table 5) require the knowledge of true graph ($\mathcal{G}$) as part of the input context to Cond-FiP. In this section we conduct where we don't provide the true graph in the input context, rather we infer the graph $\hat{\mathcal{G}}$ using an amortized causal discovery approach (AVICI (Lorch et al., 2022)) from the observational data $D_X$. We chose AVICI for this task since it can enable to amortized inference of causal graphs, hence allowing the combined pipeline of AVICI + Cond-FiP can perform amortized inference of SCMs. More precisely, AVICI infers the graph from a novel instance $\mathcal{G}$ from input context $D_X$ without updating any parameters, and we pass $(\hat{\mathcal{G}}, D_X)$ as the input context for Cond-FiP. Therefore, for any $z \in \mathbb{R}^d$, Cond-FiP ( $\mathcal{T}(z, D_X, \hat{\mathcal{G}})$) aims to replicate the functional mechanism $F(z)$ of the underlying SCM.

The results for benchmarking Cond-FiP with inferred graphs using AVICI for the task of noise prediction, sample generation, and interventional generation are provided in Table 18, Table 19, and Table 20 respectively. For a fair comparison, the baselines FiP, DECI, and DoWhy also use the inferred graph ($\hat{\mathcal{G}}$) by AVICI instead of the true graph ($\mathcal{G}$). We find that Cond-FiP remains competitive to baselines even for the scenario of unknown true causal graph. Hence, our training procedure can be extended for amortized inference of both causal graphs and causal mechanisms of the SCM.

| Method | Total Nodes | LIN **IN** | RFF **IN** | LIN **OUT** | RFF **OUT** |
|---|---|---|---|---|---|
| DoWhy | 10 | 0.16 (0.05) | 0.24 (0.04) | 0.12 (0.03) | 0.12 (0.02) |
| DECI | 10 | 0.21 (0.05) | 0.29 (0.04) | 0.16 (0.03) | 0.19 (0.04) |
| FiP | 10 | 0.16 (0.05) | 0.2 (0.04) | 0.13 (0.03) | 0.09 (0.01) |
| Cond-FiP | 10 | 0.15 (0.05) | 0.2 (0.04) | 0.13 (0.03) | 0.11 (0.01) |
| DoWhy | 20 | 0.19 (0.05) | 0.22 (0.03) | 0.2 (0.03) | 0.26 (0.01) |
| DECI | 20 | 0.23 (0.05) | 0.28 (0.03) | 0.24 (0.04) | 0.28 (0.02) |
| FiP | 20 | 0.2 (0.05) | 0.2 (0.03) | 0.21 (0.03) | 0.21 (0.02) |
| Cond-FiP | 20 | 0.18 (0.05) | 0.17 (0.02) | 0.21 (0.03) | 0.16 (0.02) |
| DoWhy | 50 | 0.44 (0.05) | 0.3 (0.03) | 0.51 (0.03) | 0.38 (0.04) |
| DECI | 50 | 0.46 (0.05) | 0.33 (0.04) | 0.52 (0.03) | 0.42 (0.05) |
| FiP | 50 | 0.44 (0.05) | 0.28 (0.04) | 0.51 (0.03) | 0.35 (0.05) |
| Cond-FiP | 50 | 0.43 (0.05) | 0.24 (0.03) | 0.53 (0.03) | 0.29 (0.04) |
| DoWhy | 100 | 0.49 (0.06) | 0.38 (0.03) | 0.64 (0.03) | 0.53 (0.04) |
| DECI | 100 | 0.5 (0.06) | 0.41 (0.03) | 0.64 (0.03) | 0.55 (0.03) |
| FiP | 100 | 0.49 (0.06) | 0.37 (0.03) | 0.64 (0.03) | 0.51 (0.04) |
| Cond-FiP | 100 | 0.48 (0.06) | 0.34 (0.03) | 0.64 (0.03) | 0.49 (0.04) |

Table 18: **Results for Noise Prediction without True Graph.** We compare Cond-FiP against the baselines for the task of predicting noise variable from input observations. Unlike experiments in Table 3, the true graph $\mathcal{G}$ is not present in input context, rather its inferred via AVICI (Lorch et al., 2022). Each cell reports the mean (standard error) RMSE over the multiple test datasets for each scenario. Shaded rows deonte the case where the graph size is larger than the train graph sizes ($d = 20$) for Cond-FiP. *Results indicate Cond-FiP can generalize to novel instances even in the absence of true graph.*

| Method | Total Nodes | LIN **IN** | RFF **IN** | LIN **OUT** | RFF **OUT** |
|---|---|---|---|---|---|
| DoWhy | 10 | 0.22 (0.07) | 0.29 (0.05) | 0.13 (0.04) | 0.14 (0.02) |
| DECI | 10 | 0.29 (0.06) | 0.39 (0.05) | 0.18 (0.04) | 0.22 (0.05) |
| FiP | 10 | 0.23 (0.06) | 0.26 (0.05) | 0.15 (0.04) | 0.12 (0.02) |
| Cond-FiP | 10 | 0.22 (0.07) | 0.26 (0.05) | 0.13 (0.04) | 0.11 (0.02) |
| DoWhy | 20 | 0.25 (0.05) | 0.38 (0.06) | 0.29 (0.06) | 0.42 (0.03) |
| DECI | 20 | 0.3 (0.06) | 0.52 (0.07) | 0.34 (0.06) | 0.47 (0.04) |
| FiP | 20 | 0.26 (0.05) | 0.37 (0.07) | 0.3 (0.06) | 0.33 (0.04) |
| Cond-FiP | 20 | 0.24 (0.05) | 0.36 (0.06) | 0.29 (0.06) | 0.35 (0.03) |
| DoWhy | 50 | 0.53 (0.07) | 0.46 (0.06) | 0.58 (0.03) | 0.59 (0.07) |
| DECI | 50 | 0.55 (0.07) | 0.54 (0.07) | 0.59 (0.02) | 0.66 (0.06) |
| FiP | 50 | 0.53 (0.07) | 0.44 (0.05) | 0.58 (0.02) | 0.53 (0.07) |
| Cond-FiP | 50 | 0.52 (0.07) | 0.43 (0.05) | 0.58 (0.02) | 0.53 (0.07) |
| DoWhy | 100 | 0.67 (0.07) | 0.52 (0.06) | 0.69 (0.02) | 0.68 (0.04) |
| DECI | 100 | 0.69 (0.08) | 0.57 (0.08) | 0.69 (0.02) | 0.71 (0.04) |
| FiP | 100 | 0.66 (0.07) | 0.5 (0.07) | 0.68 (0.02) | 0.64 (0.05) |
| Cond-FiP | 100 | 0.64 (0.06) | 0.49 (0.06) | 0.68 (0.02) | 0.63 (0.05) |

Table 19: **Results for Sample Generation without True Graph.** We compare Cond-FiP against the baselines for the task of generating samples from the input noise variable. Unlike experiments in Table 4, the true graph $\mathcal{G}$ is not present in input context, rather its inferred via AVICI (Lorch et al., 2022).. Each cell reports the mean (standard error) RMSE over the multiple test datasets for each scenario. Shaded rows deonte the case where the graph size is larger than the train graph sizes ($d = 20$) for Cond-FiP. *Results indicate Cond-FiP can generalize to novel instances even in the absence of true graph.*

| Method | Total Nodes | LIN **IN** | RFF **IN** | LIN **OUT** | RFF **OUT** |
|---|---|---|---|---|---|
| DoWhy | 10 | 0.32 (0.09) | 0.3 (0.05) | 0.13 (0.04) | 0.13 (0.02) |
| DECI | 10 | 0.37 (0.08) | 0.39 (0.05) | 0.17 (0.03) | 0.21 (0.04) |
| FiP | 10 | 0.32 (0.08) | 0.27 (0.05) | 0.14 (0.04) | 0.1 (0.02) |
| Cond-FiP | 10 | 0.31 (0.08) | 0.3 (0.05) | 0.14 (0.04) | 0.13 (0.02) |
| DoWhy | 20 | 0.29 (0.06) | 0.38 (0.07) | 0.37 (0.05) | 0.4 (0.03) |
| DECI | 20 | 0.34 (0.06) | 0.51 (0.07) | 0.41 (0.05) | 0.43 (0.03) |
| FiP | 20 | 0.3 (0.06) | 0.37 (0.07) | 0.38 (0.05) | 0.31 (0.03) |
| Cond-FiP | 20 | 0.29 (0.06) | 0.37 (0.06) | 0.37 (0.05) | 0.33 (0.03) |
| DoWhy | 50 | 0.54 (0.08) | 0.45 (0.06) | 0.62 (0.04) | 0.57 (0.06) |
| DECI | 50 | 0.57 (0.08) | 0.52 (0.07) | 0.63 (0.03) | 0.64 (0.06) |
| FiP | 50 | 0.55 (0.08) | 0.43 (0.05) | 0.62 (0.03) | 0.51 (0.07) |
| Cond-FiP | 50 | 0.54 (0.08) | 0.43 (0.05) | 0.62 (0.03) | 0.51 (0.06) |
| DoWhy | 100 | 0.66 (0.06) | 0.52 (0.07) | 0.71 (0.05) | 0.65 (0.05) |
| DECI | 100 | 0.68 (0.07) | 0.58 (0.09) | 0.71 (0.05) | 0.7 (0.04) |
| FiP | 100 | 0.65 (0.06) | 0.51 (0.07) | 0.71 (0.05) | 0.62 (0.05) |
| Cond-FiP | 100 | 0.64 (0.06) | 0.49 (0.06) | 0.7 (0.04) | 0.62 (0.05) |

Table 20: **Results for Interventional Generation without True Graph.** We compare Cond-FiP against the baselines for the task of interventional data from the input noise variable. Unlike experiments in Table 5, the true graph $\mathcal{G}$ is not present in input context, rather its inferred via AVICI (Lorch et al., 2022). Each cell reports the mean (standard error) RMSE over the multiple test datasets for each scenario. Shaded rows deonte the case where the graph size is larger than the train graph sizes ($d = 20$) for Cond-FiP. *Results indicate Cond-FiP can generalize to novel instances even in the absence of true graph.*

### F.1 Analyzing Cond-FiP's Sensitivity to Causal Graph Structure

To better understand the ability of Cond-FiP to capture graph-specific structure, we conduct a systematic analysis by introducing random perturbations ($p$) to the true causal graph. Specifically, we randomly remove a proportion $p$ of the true edges, such that on average $p \times$ total edges are missing. We report results for sample generation in the challenging OOD setting ($d{=}100$ & RFF **OUT**).

Our findings (Table 21) show that Cond-FiP's performance shows significant difference after we remove 5 edges ($p = 0.05$). Across all tested levels of perturbation, Cond-FiP is competitive with FiP, a baseline trained from scratch for each scenario. These results demonstrate that Cond-FiP exhibits robustness to moderate errors in the input causal graph, comparable to a baseline trained from scratch. This provides more evidence to our claim that Cond-FiP learns causal mechanisms during training and can adapt to new contexts at test time by inferring functions that best explain the available information in the context (input graph and observations), even when the input causal graph is inaccurate.

|          | $p = 0$     | $p = 0.01$  | $p = 0.02$  | $p = 0.05$  | $p = 0.1$   |
|----------|-------------|-------------|-------------|-------------|-------------|
| FiP      | 0.55 (0.08) | 0.55 (0.08) | 0.57 (0.08) | 0.62 (0.08) | 0.68 (0.08) |
| Cond-FiP | 0.57 (0.07) | 0.58 (0.07) | 0.59 (0.07) | 0.62 (0.07) | 0.67 (0.07) |

Table 21: **Robustness to causal graph errors.** We compare Cond-FiP for varying levels of corruption ($p$) to the true causal graph by randomly removing a proportion $p$ of edges. Results show that Cond-FiP remains competitive with FiP, even as the input graph becomes increasingly inaccurate, demonstrating robustness to moderate structural errors and its ability to utilize the given imperfect causal information.

# G   Ablation Study on AVICI benchmark

## G.1   Analyzing the Effect of Pretraining Scale

To better understand the scaling of Cond-FiP w.r.t the pretraining data, we conduct experiments at smaller scales, with a total of $1e5, 4e5, 1e6$ SCMs, as opposed to using $4e6$ SCMs in our main body results.

Table 22 reports sample-generation accuracy for the $d = 20$ setting. As expected, Cond-FiP benefits consistently from additional pretraining data: all four metrics (LIN **IN**, RFF **IN**, LIN **OUT**, RFF **OUT**) improve monotonically with scale. The most pronounced gains appear when increasing the pretraining size from $1e5$ to $4e5$, while improvements become more incremental beyond $1e6$. These results highlight that Cond-FiP continues to leverage larger synthetic datasets, though the returns gradually diminish as scale increases.

| Method | Total Nodes | LIN **IN** | RFF **IN** | LIN **OUT** | RFF **OUT** |
|---|---|---|---|---|---|
| Cond-FiP($1e5$) | 20 | 0.11 (0.01) | 0.32 (0.06) | 0.16 (0.03) | 0.44 (0.04) |
| Cond-FiP($4e5$) | 20 | 0.07 (0.01) | 0.28 (0.06) | 0.11 (0.01) | 0.37 (0.03) |
| Cond-FiP($1e6$) | 20 | 0.07 (0.01) | 0.25 (0.06) | 0.09 (0.01) | 0.33 (0.03) |
| Cond-FiP($4e6$) | 20 | 0.05 (0.01) | 0.24 (0.06) | 0.07 (0.01) | 0.30 (0.03) |

Table 22: **Effect of Pretraining Scale.** Sample generation performance of Cond-FiP ($d = 20$) under varying amounts of pretraining data. Larger synthetic SCM pretraining datasets lead to consistent improvements across all scenarios, with diminishing but still positive gains beyond $1e6$ SCMs.

## G.2   Ablation Study of Encoder

We conduct an ablation study where we train two variants of the encoder in Cond-FiP described as follows:

- *Cond-FiP (LIN)*: We sample SCMs with linear causal mechanisms during training of the encoder.

- *Cond-FiP (RFF)*: We sample SCMs with non-linear causal mechanisms during training of the encoder.

Note that for the training the subsequent decoder, we sample SCMs with both linear and rff causal mechanisms as in the main results ( Table 3, Table 4, and Table 5). Note that in the main results, the encoder was trained by sampling SCMs with both linear and rff functional relationships. Hence, this ablation helps us to understand whether the strategy of training encoder on mixed functional relationships can bring more generalization to the amortization process, or if we should have trained encoders specialized for linear and non-linear functional relationships.

We present our results of the ablation study for the task of noise prediction, sample generation, and interventional generation in Table 23, Table 24, Table 25 respectively. Our findings indicate that Cond-FiP is robust to the choice of encoder training strategy! Even though the encoder for Cond-FiP (RFF) was only trained on data from non-linear SCMs, its generalization performance is similar to Cond-FiP where the encoder was trained on data from both linear and non-linear SCMs.

| Method | Total Nodes | LIN **IN** | RFF **IN** | LIN **OUT** | RFF **OUT** |
|---|---|---|---|---|---|
| Cond-FiP(LIN) | 10 | 0.07 (0.01) | 0.21 (0.02) | 0.08 (0.01) | 0.2 (0.03) |
| Cond-FiP(RFF) | 10 | 0.06 (0.01) | 0.11 (0.01) | 0.07 (0.01) | 0.09 (0.01) |
| Cond-FiP | 10 | 0.06 (0.01) | 0.1 (0.01) | 0.07 (0.01) | 0.1 (0.01) |
| Cond-FiP(LIN) | 20 | 0.07 (0.01) | 0.19 (0.02) | 0.09 (0.01) | 0.21 (0.01) |
| Cond-FiP(RFF) | 20 | 0.06 (0.01) | 0.09 (0.01) | 0.1 (0.02) | 0.11 (0.01) |
| Cond-FiP | 20 | 0.06 (0.01) | 0.09 (0.01) | 0.07 (0.0) | 0.12 (0.0) |
| Cond-FiP(LIN) | 50 | 0.07 (0.01) | 0.21 (0.02) | 0.07 (0.01) | 0.24 (0.01) |
| Cond-FiP(RFF) | 50 | 0.07 (0.01) | 0.09 (0.01) | 0.07 (0.10) | 0.14 (0.01) |
| Cond-FiP | 50 | 0.06 (0.01) | 0.1 (0.01) | 0.07 (0.01) | 0.14 (0.01) |
| Cond-FiP(LIN) | 100 | 0.06 (0.0) | 0.22 (0.02) | 0.07 (0.01) | 0.26 (0.01) |
| Cond-FiP(RFF) | 100 | 0.06 (0.01) | 0.09 (0.01) | 0.07 (0.01) | 0.14 (0.01) |
| Cond-FiP | 100 | 0.05 (0.0) | 0.1 (0.01) | 0.07 (0.01) | 0.16 (0.01) |

Table 23: **Encoder Ablation for Noise Prediction.** We compare Cond-FiP against the baselines for the task of predicting noise variable from input observations against two variants. One variant corresponds to the encoder trained on SCMs with only linear functional relationships, Cond-FiP(LIN). Similarly, we have another variant where the decoder was trained on SCMs with only rff functional relationships, Cond-FiP(RFF). Each cell reports the mean (standard error) RMSE over the multiple test datasets for each scenario. *Results show that training on only non-linear SCMs (*Cond-FiP(RFF)*) gives similar performance as training on both linear and non-linear SCMs (*Cond-FiP*).*

| Method | Total Nodes | LIN **IN** | RFF **IN** | LIN **OUT** | RFF **OUT** |
|---|---|---|---|---|---|
| Cond-FiP(LIN) | 10 | 0.05 (0.01) | 0.14 (0.02) | 0.06 (0.0) | 0.08 (0.01) |
| Cond-FiP(RFF) | 10 | 0.08 (0.01) | 0.18 (0.06) | 0.06 (0.0) | 0.07 (0.01) |
| Cond-FiP | 10 | 0.06 (0.01) | 0.14 (0.02) | 0.05 (0.01) | 0.08 (0.01) |
| Cond-FiP(LIN) | 20 | 0.05 (0.01) | 0.25 (0.06) | 0.07 (0.01) | 0.3 (0.03) |
| Cond-FiP(RFF) | 20 | 0.08 (0.01) | 0.22 (0.05) | 0.11 (0.01) | 0.29 (0.03) |
| Cond-FiP | 20 | 0.05 (0.01) | 0.24 (0.06) | 0.07 (0.01) | 0.3 (0.03) |
| Cond-FiP(LIN) | 50 | 0.08 (0.01) | 0.26 (0.05) | 0.11 (0.04) | 0.52 (0.08) |
| Cond-FiP(RFF) | 50 | 0.11 (0.01) | 0.26 (0.05) | 0.15 (0.02) | 0.48 (0.07) |
| Cond-FiP | 50 | 0.08 (0.01) | 0.25 (0.05) | 0.07 (0.0) | 0.48 (0.07) |
| Cond-FiP(LIN) | 100 | 0.07 (0.01) | 0.27 (0.06) | 0.08 (0.0) | 0.57 (0.07) |
| Cond-FiP(RFF) | 100 | 0.11 (0.01) | 0.29 (0.08) | 0.18 (0.03) | 0.61 (0.08) |
| Cond-FiP | 100 | 0.07 (0.01) | 0.29 (0.07) | 0.09 (0.01) | 0.57 (0.07) |

Table 24: **Encoder Ablation for Sample Generation.** We compare Cond-FiP against the baselines for the task of generating samples from input noise variables against two variants. One variant corresponds to the encoder trained on SCMs with only linear functional relationships, Cond-FiP(LIN). Similarly, we have another variant where the decoder was trained on SCMs with only rff functional relationships, Cond-FiP(RFF). Each cell reports the mean (standard error) RMSE over the multiple test datasets for each scenario. *Results show that training on only non-linear SCMs (*Cond-FiP(RFF)*) gives similar performance as training on both linear and non-linear SCMs (*Cond-FiP*).*

| Method | Total Nodes | LIN **IN** | RFF **IN** | LIN **OUT** | RFF **OUT** |
|---|---|---|---|---|---|
| Cond-FiP(LIN) | 10 | 0.09 (0.02) | 0.2 (0.03) | 0.06 (0.01) | 0.1 (0.01) |
| Cond-FiP(RFF) | 10 | 0.13 (0.04) | 0.23 (0.08) | 0.08 (0.01) | 0.1 (0.01) |
| Cond-FiP | 10 | 0.1 (0.03) | 0.21 (0.03) | 0.07 (0.01) | 0.11 (0.01) |
| Cond-FiP(LIN) | 20 | 0.08 (0.01) | 0.24 (0.05) | 0.12 (0.04) | 0.3 (0.03) |
| Cond-FiP(RFF) | 20 | 0.13 (0.02) | 0.23 (0.05) | 0.13 (0.03) | 0.31 (0.02) |
| Cond-FiP | 20 | 0.09 (0.01) | 0.24 (0.05) | 0.14 (0.03) | 0.31 (0.03) |
| Cond-FiP(LIN) | 50 | 0.12 (0.02) | 0.29 (0.05) | 0.1 (0.01) | 0.51 (0.07) |
| Cond-FiP(RFF) | 50 | 0.14 (0.02) | 0.29 (0.05) | 0.18 (0.03) | 0.47 (0.06) |
| Cond-FiP | 50 | 0.13 (0.02) | 0.27 (0.04) | 0.12 (0.02) | 0.48 (0.07) |
| Cond-FiP(LIN) | 100 | 0.1 (0.01) | 0.3 (0.06) | 0.12 (0.01) | 0.56 (0.07) |
| Cond-FiP(RFF) | 100 | 0.12 (0.01) | 0.31 (0.07) | 0.2 (0.04) | 0.6 (0.09) |
| Cond-FiP | 100 | 0.1 (0.01) | 0.3 (0.06) | 0.14 (0.02) | 0.58 (0.07) |

Table 25: **Encoder Ablation for Interventional Generation.** We compare Cond-FiP against the baselines for the task of generating interventional data from input noise variables against two variants. One variant corresponds to the encoder trained on SCMs with only linear functional relationships, Cond-FiP(LIN). Similarly, we have another variant where the decoder was trained on SCMs with only rff functional relationships, Cond-FiP(RFF). Each cell reports the mean (standard error) RMSE over the multiple test datasets for each scenario. *Results show that training on only non-linear SCMs (*Cond-FiP(RFF)*) gives similar performance as training on both linear and non-linear SCMs (*Cond-FiP*).*

### G.3 Ablation Study of Decoder

We conduct an ablation study where we train two variants of the decoder Cond-FiP described as follows:

- *Cond-FiP (LIN):* We sample SCMs with linear functional relationships during training.

- *Cond-FiP (RFF):* We sample SCMs with non-linear functional relationships for training.

Note that in the main results (Table 4, Table 5) we show the performances of Cond-FiP trained by sampling SCMs with both linear and non-linear causal mechanisms. Hence, this ablations helps us to understand whether the strategy of training on mixed causal mechanisms can bring more generalization to the amortization process, or if we should have trained decoders specialized for linear and non-linear functional relationships.

We present the results of our ablation study in Table 26 and Table 27, for the task of sample generation and interventional generation respectively. Our findings indicate that Cond-FiP decoder trained for both linear and non-linear functional relationships is able to specialize for both the scenarios. While Cond-FiP (LIN) is only able to perform well for linear benchmarks, and similarly Cond-FiP (RFF) can only achieve decent predictions for non-linear benchmarks, Cond-FiP is achieve the best performances on both the linear and non-linear benchmarks.

| Method | Total Nodes | LIN **IN** | RFF **IN** | LIN **OUT** | RFF **OUT** |
|---|---|---|---|---|---|
| Cond-FiP(LIN) | 10 | 0.07 (0.02) | 0.4 (0.06) | 0.07 (0.01) | 0.25 (0.06) |
| Cond-FiP(RFF) | 10 | 0.1 (0.02) | 0.15 (0.02) | 0.08 (0.01) | 0.09 (0.01) |
| Cond-FiP | 10 | 0.06 (0.01) | 0.14 (0.02) | 0.05 (0.01) | 0.08 (0.01) |
| Cond-FiP(LIN) | 20 | 0.07 (0.01) | 0.44 (0.07) | 0.10 (0.01) | 0.58 (0.02) |
| Cond-FiP(RFF) | 20 | 0.11 (0.01) | 0.26 (0.06) | 0.14 (0.01) | 0.31 (0.03) |
| Cond-FiP | 20 | 0.05 (0.01) | 0.24 (0.06) | 0.07 (0.01) | 0.3 (0.03) |
| Cond-FiP(LIN) | 50 | 0.10 (0.01) | 0.5 (0.07) | 0.14 (0.02) | 0.69 (0.04) |
| Cond-FiP(RFF) | 50 | 0.15 (0.02) | 0.27 (0.05) | 0.19 (0.02) | 0.5 (0.07) |
| Cond-FiP | 50 | 0.08 (0.01) | 0.25 (0.05) | 0.07 (0.0) | 0.48 (0.07) |
| Cond-FiP(LIN) | 100 | 0.1 (0.01) | 0.51 (0.07) | 0.15 (0.02) | 0.72 (0.04) |
| Cond-FiP(RFF) | 100 | 0.16 (0.03) | 0.29 (0.07) | 0.27 (0.04) | 0.59 (0.06) |
| Cond-FiP | 100 | 0.07 (0.01) | 0.29 (0.07) | 0.09 (0.01) | 0.57 (0.07) |

Table 26: **Decoder Ablation for Sample Generation.** We compare Cond-FiP for the task of generating samples from input noise variables against two variants. One variant corresponds to a decoder trained on SCMs with only linear functional relationships, Cond-FiP(LIN). Similarly, we have another variant where the decoder was trained on SCMs with only rff functional relationships, Cond-FiP(RFF). Each cell reports the mean (standard error) RMSE over the multiple test datasets for each scenario. *Results indicate that training on both linear and non-linear SCMs is crucial to generalize effectively in all scenarios.*

| Method | Total Nodes | LIN **IN** | RFF **IN** | LIN **OUT** | RFF **OUT** |
|---|---|---|---|---|---|
| Cond-FiP(LIN) | 10 | 0.09 (0.02) | 0.40 (0.07) | 0.06 (0.01) | 0.22 (0.04) |
| Cond-FiP(RFF) | 10 | 0.16 (0.05) | 0.22 (0.03) | 0.08 (0.01) | 0.11 (0.01) |
| Cond-FiP | 10 | 0.10 (0.03) | 0.21 (0.03) | 0.07 (0.01) | 0.11 (0.01) |
| Cond-FiP(LIN) | 20 | 0.10 (0.01) | 0.45 (0.07) | 0.16 (0.03) | 0.57 (0.02) |
| Cond-FiP(RFF) | 20 | 0.14 (0.02) | 0.26 (0.05) | 0.21 (0.03) | 0.32 (0.02) |
| Cond-FiP | 20 | 0.09 (0.01) | 0.24 (0.05) | 0.14 (0.03) | 0.31 (0.03) |
| Cond-FiP(LIN) | 50 | 0.14 (0.02) | 0.49 (0.07) | 0.14 (0.02) | 0.68 (0.04) |
| Cond-FiP(RFF) | 50 | 0.19 (0.03) | 0.28 (0.05) | 0.21 (0.03) | 0.49 (0.06) |
| Cond-FiP | 50 | 0.13 (0.02) | 0.27 (0.04) | 0.12 (0.02) | 0.48 (0.07) |
| Cond-FiP(LIN) | 100 | 0.12 (0.02) | 0.52 (0.07) | 0.18 (0.03) | 0.71 (0.04) |
| Cond-FiP(RFF) | 100 | 0.18 (0.03) | 0.32 (0.07) | 0.24 (0.04) | 0.59 (0.07) |
| Cond-FiP | 100 | 0.10 (0.01) | 0.30 (0.06) | 0.14 (0.02) | 0.58 (0.07) |

Table 27: **Decoder Ablation for Interventional Generation.** We compare Cond-FiP against two variants for the task of interventional data from input noise variables. One variant corresponds to a decoder trained on SCMs with only linear functional relationships, Cond-FiP(LIN). Similarly, we have another variant where the decoder was trained on SCMs with only rff functional relationships, Cond-FiP(RFF). Each cell reports the mean (standard error) RMSE over the multiple test datasets for each scenario. *Results indicate that training on both linear and non-linear SCMs is crucial to generalize effectively in all scenarios.*

# H   Comparing Cond-FiP with CausalNF

We also compare Cond-FiP with CausalNF (Javaloy et al., 2023) for the task of noise prediction (Table 28) and sample generation (Table 29). The test datasets consist of $n_{\text{test}} = 400$ samples, exact same setup as in our main results (Table 3, Table 4, and Table 5). To ensure a fair comparison, we provided CausalNF with the true causal graph.

Our analysis reveals that CausalNF underperforms compared to Cond-FiP in both tasks, and it is also a weaker baseline relative to FiP. Note also the authors did not experiment with large graphs for CausalNF; the largest graph they used contained approximately 10 nodes. Also, they trained CausalNF on much larger datasets with a sample size of 20k, while our setup has datasets with 400 samples only.

| Method | Total Nodes | LIN **IN** | RFF **IN** | LIN **OUT** | RFF **OUT** |
|---|---|---|---|---|---|
| CausalNF | 10 | 0.16 (0.02) | 0.41 (0.09) | 0.38 (0.04) | 0.35 (0.02) |
| Cond-FiP | 10 | 0.06 (0.01) | 0.10 (0.01) | 0.07 (0.01) | 0.10 (0.01) |
| CausalNF | 20 | 0.18 (0.03) | 0.45 (0.12) | 0.29 (0.05) | 0.36 (0.03) |
| Cond-FiP | 20 | 0.06 (0.01) | 0.09 (0.01) | 0.07 (0.00) | 0.12 (0.00) |
| CausalNF | 50 | 0.25 (0.03) | 0.56 (0.09) | 0.45 (0.06) | 0.38 (0.04) |
| Cond-FiP | 50 | 0.06 (0.01) | 0.10 (0.01) | 0.07 (0.01) | 0.14 (0.01) |
| CausalNF | 100 | 0.24 (0.02) | 0.80 (0.1) | 0.37 (0.06) | 0.49 (0.05) |
| Cond-FiP | 100 | 0.05 (0.0) | 0.10 (0.01) | 0.07 (0.01) | 0.16 (0.01) |

Table 28: **Results for Noise Prediction with CausalNF.** We compare Cond-FiP against CausalNF for the task of predicting noise variables from input observations. *We find that CausalNF underperforms compared to Cond-FiP by a significant margin.*

| Method | Total Nodes | LIN **IN** | RFF **IN** | LIN **OUT** | RFF **OUT** |
|---|---|---|---|---|---|
| CausalNF | 10 | 0.27 (0.07) | 0.29 (0.04) | 0.20 (0.03) | 0.20 (0.03) |
| Cond-FiP | 10 | 0.06 (0.01) | 0.14 (0.02) | 0.05 (0.01) | 0.08 (0.01) |
| CausalNF | 20 | 0.23 (0.02) | 0.36 (0.05) | 0.22 (0.02) | 0.45 (0.02) |
| Cond-FiP | 20 | 0.05 (0.01) | 0.24 (0.06) | 0.07 (0.01) | 0.30 (0.03) |
| CausalNF | 50 | 1.5 (0.26) | 0.93 (0.13) | 3.09 (0.55) | 0.95 (0.04) |
| Cond-FiP | 50 | 0.08 (0.01) | 0.25 (0.05) | 0.07 (0.00) | 0.48 (0.07) |
| CausalNF | 100 | 1.23 (0.13) | 0.85 (0.08) | 1.67 (0.13) | 0.96 (0.04) |
| Cond-FiP | 100 | 0.07 (0.01) | 0.29 (0.07) | 0.09 (0.01) | 0.57 (0.07) |

Table 29: **Results for Sample Generation with CausalNF.** We compare Cond-FiP against CausalNF for the task of generating samples from input noise variables. *We find that CausalNF underperforms compared to Cond-FiP by a significant margin.*

# I   Limitations of Cond-FiP

## I.1   Evaluating Generalization of Cond-Fip to Larger Sample Size

In the main results (Table 3, Table 4, and Table 5), we evaluated Cond-FiP's generalization capabilities to larger graphs ($d = 50$, $d = 100$) than those used for training ($d = 20$). In this section, we carry a similar experiment where instead of increasing the total nodes in the graph, we test Cond-FiP on datasets with more samples $n_{\mathcal{D}_{\text{test}}} = 1000$, while Cond-FiP was only trained for datasets with sample size $n_{\mathcal{D}} = 400$.

The results for the experiments are presented in Table 30, Table 31, and Table 32 for the task of noise prediction, sample generation, and interventional generation respectively. Our findings indicate that Cond-FiP is still able to compete with other baseline in this regime. However, we observe that the performances of Cond-FiP did not improve by increasing the sample size compared to the results obtained for the 400 samples case, meaning that the performance of our models depends exclusively on the setting used at training time. We leave for future works the learning of a larger instance of Cond-FiP trained on larger sample size problems.

| Method | Total Nodes | LIN **IN** | RFF **IN** | LIN **OUT** | RFF **OUT** |
|---|---|---|---|---|---|
| DoWhy | 10 | 0.02 (0.0) | 0.10 (0.01) | 0.21 (0.04) | 0.23 (0.02) |
| DECI | 10 | 0.05 (0.01) | 0.12 (0.01) | 0.21 (0.04) | 0.27 (0.03) |
| FiP | 10 | 0.03 (0.0) | 0.06 (0.0) | 0.21 (0.04) | 0.23 (0.02) |
| Cond-FiP | 10 | 0.05 (0.01) | 0.11 (0.01) | 0.21 (0.04) | 0.25 (0.02) |
| DoWhy | 20 | 0.02 (0.0) | 0.11 (0.02) | 0.16 (0.01) | 0.3 (0.02) |
| DECI | 20 | 0.04 (0.01) | 0.11 (0.02) | 0.16 (0.01) | 0.29 (0.02) |
| FiP | 20 | 0.03 (0.0) | 0.08 (0.02) | 0.16 (0.01) | 0.26 (0.02) |
| Cond-FiP | 20 | 0.06 (0.01) | 0.09 (0.01) | 0.18 (0.01) | 0.26 (0.01) |

Table 30: **Results for Noise Prediction with Larger Sample Size ($n_{\mathcal{D}_{\text{test}}} = 1000$).** We compare Cond-FiP against the baselines for the task of predicting noise variables from the input observations. Each cell reports the mean (standard error) RMSE over the multiple test datasets for each scenario. *Results indicate that Cond-FiP does not yet benefit from larger context sizes at inference, suggesting the need to scale both the model and training data for richer contexts.*

| Method | Total Nodes | LIN **IN** | RFF **IN** | LIN **OUT** | RFF **OUT** |
|---|---|---|---|---|---|
| DoWhy | 10 | 0.04 (0.0) | 0.14 (0.02) | 0.29 (0.04) | 0.3 (0.03) |
| DECI | 10 | 0.07 (0.01) | 0.17 (0.02) | 0.29 (0.04) | 0.33 (0.04) |
| FiP | 10 | 0.05 (0.0) | 0.09 (0.01) | 0.29 (0.04) | 0.29 (0.03) |
| Cond-FiP | 10 | 0.05 (0.01) | 0.14 (0.02) | 0.29 (0.04) | 0.29 (0.03) |
| DoWhy | 20 | 0.04 (0.01) | 0.21 (0.05) | 0.28 (0.01) | 0.55 (0.06) |
| DECI | 20 | 0.07 (0.01) | 0.21 (0.04) | 0.29 (0.01) | 0.59 (0.06) |
| FiP | 20 | 0.05 (0.0) | 0.17 (0.04) | 0.28 (0.01) | 0.53 (0.06) |
| Cond-FiP | 20 | 0.05 (0.0) | 0.24 (0.05) | 0.28 (0.01) | 0.53 (0.06) |

Table 31: **Results for Sample Generation with Larger Sample Size ($n_{\mathcal{D}_{\text{test}}} = 1000$).** We compare Cond-FiP against the baselines for the task of generating samples from the input noise variables. Each cell reports the mean (standard error) RMSE over the multiple test datasets for each scenario. *Results indicate that Cond-FiP does not yet benefit from larger context sizes at inference, suggesting the need to scale both the model and training data for richer contexts.*

| Method | Total Nodes | LIN **IN** | RFF **IN** | LIN **OUT** | RFF **OUT** |
|---|---|---|---|---|---|
| DoWhy | 10 | 0.04 (0.01) | 0.16 (0.03) | 0.26 (0.03) | 0.27 (0.03) |
| DECI | 10 | 0.09 (0.01) | 0.19 (0.02) | 0.26 (0.03) | 0.31 (0.04) |
| FiP | 10 | 0.05 (0.01) | 0.12 (0.02) | 0.26 (0.03) | 0.27 (0.03) |
| Cond-FiP | 10 | 0.09 (0.02) | 0.19 (0.03) | 0.27 (0.03) | 0.3 (0.03) |
| DoWhy | 20 | 0.04 (0.0) | 0.20 (0.04) | 0.26 (0.01) | 0.53 (0.06) |
| DECI | 20 | 0.08 (0.01) | 0.20 (0.03) | 0.29 (0.02) | 0.54 (0.05) |
| FiP | 20 | 0.06 (0.01) | 0.16 (0.04) | 0.28 (0.02) | 0.48 (0.06) |
| Cond-FiP | 20 | 0.07 (0.01) | 0.27 (0.05) | 0.30 (0.02) | 0.51 (0.06) |

Table 32: **Results for Interventional Generation with Larger Sample Size ($n_{\mathcal{D}_{\text{test}}} = 1000$).** We compare Cond-FiP against the baselines for the task of generating interventional data from the input noise variables. Each cell reports the mean (standard error) RMSE over the multiple test datasets for each scenario. *Results indicate that Cond-FiP does not yet benefit from larger context sizes at inference, suggesting the need to scale both the model and training data for richer contexts.*

### I.2 Counterfactual Generation with Cond-FiP

We provide results (Table 33) for bechmarking Cond-FiP against baselines for the task of counterfactual generation. We operate in the same setup as the one in our main results ($n_{\mathcal{D}_{\text{test}}} = 400$) Appendix C and all the methods are provided with the true casual graph. We observe that Unlike the tasks of noise prediction, sample & interventional generation, we find that Cond-FiP is worse than the baselines for the task of counterfactual generation. This can be explained as the training of Cond-FiP decoder relies on the true noise variables, and the model struggles to generalize the learned functional mechanisms when provided with inferred noise variables. We leave the improvement of Cond-FiP for counterfactual generation as future work.

| Method | Total Nodes | LIN **IN** | RFF **IN** | LIN **OUT** | RFF **OUT** |
|---|---|---|---|---|---|
| DoWhy | 10 | 0.03 (0.03) | 0.13 (0.03) | 0.0 (0.0) | 0.04 (0.01) |
| DECI | 10 | 0.1 (0.02) | 0.2 (0.03) | 0.04 (0.01) | 0.11 (0.02) |
| FiP | 10 | 0.03 (0.01) | 0.09 (0.02) | 0.02 (0.0) | 0.03 (0.01) |
| Cond-FiP | 10 | 0.09 (0.03) | 0.21 (0.03) | 0.05 (0.01) | 0.11 (0.01) |
| DoWhy | 20 | 0.01 (0.0) | 0.12 (0.03) | 0.0 (0.0) | 0.13 (0.02) |
| DECI | 20 | 0.06 (0.01) | 0.15 (0.03) | 0.07 (0.03) | 0.15 (0.02) |
| FiP | 20 | 0.03 (0.01) | 0.1 (0.03) | 0.06 (0.04) | 0.09 (0.02) |
| Cond-FiP | 20 | 0.09 (0.02) | 0.26 (0.05) | 0.13 (0.02) | 0.3 (0.03) |
| DoWhy | 50 | 0.0 (0.0) | 0.09 (0.02) | 0.0 (0.0) | 0.17 (0.04) |
| DECI | 50 | 0.04 (0.01) | 0.11 (0.02) | 0.03 (0.01) | 0.18 (0.04) |
| FiP | 50 | 0.03 (0.01) | 0.08 (0.02) | 0.03 (0.01) | 0.14 (0.04) |
| Cond-FiP | 50 | 0.1 (0.02) | 0.26 (0.04) | 0.1 (0.01) | 0.46 (0.06) |
| DoWhy | 100 | 0.0 (0.0) | 0.08 (0.02) | 0.0 (0.0) | 0.2 (0.05) |
| DECI | 100 | 0.02 (0.01) | 0.1 (0.02) | 0.02 (0.01) | 0.22 (0.05) |
| FiP | 100 | 0.01 (0.01) | 0.07 (0.02) | 0.02 (0.01) | 0.19 (0.05) |
| Cond-FiP | 100 | 0.09 (0.02) | 0.29 (0.06) | 0.13 (0.02) | 0.56 (0.08) |

Table 33: **Results for Counterfactual Generation.** We compare Cond-FiP against the baselines for the task of generating counterfactual data from the input noise variables. Each cell reports the mean (standard error) RMSE over the multiple test datasets for each scenario. Shaded rows denote the case where the graph size is larger than the train graph sizes ($d = 20$) for Cond-FiP. *Results indicate that Cond-FiP struggles with counterfactual generation and cannot always match the performance of baselines trained from scratch.*

