# OpenReview forum: "Amortized Inference of Causal Models via Conditional Fixed-Point Iterations"
_TMLR — Accepted by TMLR_

### Review · Reviewer_1ybH · 2025-10-30

**Summary Of Contributions:**

The paper tackles the problem of learning SCMs in an amortized fashion, aiming to infer causal mechanisms across multiple datasets using a single, reusable model. The authors introduce Cond-FiP, a conditional extension of the recently proposed Fixed-Point (FiP) approach for causal generative modeling. While FiP must be trained separately for each dataset, Cond-FiP incorporates dataset-conditioned embeddings that enable amortized inference of causal mechanisms, allowing the same model to adapt to unseen SCMs without further training.

The proposed framework consists of two components:
1) a dataset encoder that learns context embeddings from observational data and causal graphs using transformer-based in-context learning
2) a conditional decoder that models causal mechanisms as conditional fixed-point iterations, integrating the dataset embeddings through adaptive layer normalization and context-dependent codebooks.
This setup allows Cond-FiP to reconstruct noise variables, generate both observational and interventional data, and generalize to unseen causal models

Empirically, the authors evaluate Cond-FiP on a large suite of synthetic and real-world benchmarks (including AVICI, C-Suite, and the Sachs dataset). The results demonstrate that Cond-FiP achieves performance comparable to or better than per-dataset baselines (FiP, DECI, and DoWhy) while generalizing across out-of-distribution settings, varying graph sizes, unknown causal graphs, and limited data scenarios. The model also shows robustness when applied to unseen data simulators and complex noise distributions.

Key strengths:
* Introduces a novel amortized inference framework for SCMs, addressing a significant limitation of existing methods that require retraining per dataset.
* Conceptually elegant integration of dataset conditioning into the FiP architecture through adaptive normalization and context-dependent embeddings.
* Thorough experimental evaluation across diverse and challenging settings, including tests of generalization, robustness to distribution shifts, and transfer to real-world data.
* Clear presentation and strong empirical evidence supporting the claims of improved data efficiency and generalization.

Key weaknesses (minor at this stage):
* The approach relies on the additive noise model assumption during training, limiting applicability to more general classes of SCMs.
* The encoder requires access to true noise variables, which restricts end-to-end training on real-world data.
* Experiments focus mostly on synthetic data, and real-world validation remains limited in scope.

**Audience:**

Yes

**Audience Explanation:**

The paper's topic is highly relevant to the TMLR audience, particularly those interested in causal machine learning, meta-learning, and generative modeling. The proposed method addresses a key limitation in current causal generative modeling: the need to train separate models for each dataset. By introducing a conditional, amortized approach, the paper connects ideas from causal inference, transformer-based in-context learning, and meta-learning, three areas that are currently of significant interest in the community.

Moreover, the work contributes to an emerging research direction on “causal foundation models,” which aim to generalize causal reasoning across tasks. The presented method offers a concrete, well-evaluated step in that direction, supported by extensive empirical evidence.

**Broader Impact Concerns:**

The paper includes a Broader Impact Statement, and it is generally appropriate for a technical contribution of this nature. The work focuses on a methodological advance without targeting any specific downstream application domain. The authors correctly state that the research does not pose foreseeable societal risks in its current form.

**Claims And Evidence:**

Yes

**Claims Explanation:**

The paper’s central assertion of Cond-FiP enabling amortized inference of causal mechanisms across different SCMs without retraining is validated through a comprehensive set of experiments. The authors systematically compare Cond-FiP against strong baselines (FiP, DECI, DoWhy) across multiple benchmarks and evaluation tasks (noise prediction, sample generation, and interventional generation).

The results consistently show that Cond-FiP performs on par with or better than models trained from scratch, both in-distribution and under substantial distribution shifts, such as unseen causal mechanisms, noise types, and larger graph sizes. Additional experiments on scarce-data settings, unknown causal graphs, and different simulators (C-Suite) provide further support for the generalization claims. The real-world Sachs dataset evaluation (while limited) also confirms that Cond-FiP matches baseline performance even without retraining.

**Requested Changes:**

Requested Changes
1) Clarify the role and limitations of the ANM assumption: The training of the dataset encoder relies on the ANM assumption and access to true noise variables, which substantially limits applicability to real-world data. The authors should clearly state this limitation in the main text (not only in the appendix or conclusion) and discuss potential strategies for relaxing this assumption, such as self-supervised or implicit encoding methods.
2) Improve the presentation of the methodology section: The description of the Cond-FiP architecture is technically detailed but dense, making it difficult to follow the flow from dataset encoding to conditional decoding. Adding a small schematic or pseudo-code block summarizing the data flow (input -> embedding -> conditional codebook -> adaptive transformer -> output) would make the method more accessible to readers less familiar with FiP.
3) Expand the real-world evaluation: The Sachs dataset experiment is a good starting point but currently limited in scope. Even if only partially feasible, including one or two additional small-scale real datasets (e.g., synthetic biology, causal discovery benchmarks) or reporting qualitative results would demonstrate the model’s broader applicability and enhance the paper’s impact.
4) Discuss computational efficiency and scaling behavior: The amortized nature of Cond-FiP is claimed to bring computational savings, but no quantitative evidence (e.g., runtime or training cost per dataset) is provided. A small comparison table summarizing training and inference time relative to FiP or DECI would highlight the practical benefits of amortization more.
5) Minor textual and formatting revisions: A few small typographical inconsistencies (e.g., “scare data regime” instead of “scarce”) should be corrected. Figures 2–4 would benefit from increasing the width (emoving the whitespace) and from using a larger font and clearer legends.

---

> ### Author Response · Authors · 2025-11-10
>
> We thank the reviewer for their positive and insightful feedback! We appreciate they found our framework novel and our experiments comprehensive and challenging. We now address the concerns raised by them below.
>
> > Clarify the role and limitations of the ANM assumption
>
> Thanks for raising this concern. In the rebuttal revision, we have added a paragraph titled "remark on ANM assumption" at the end of Section 3.2, which highlights this issue limited to training of the dataset encoder and provide future directions for alleviating it.
>
> > Improve the presentation of the methodology section
>
> We appreciate this suggestion and we have added a pseudo code in Appendix B.5 for both the training of the dataset encoder and the cond-FiP method. We hope this should improve the presentation of the methodology section.
>
> > Expand the real-world evaluation
>
> Thanks for raising this point. We have added the experiments on the ecoli dataset from the bnlearn repository in the rebuttal revision (Table 2), which shows that Cond-FiP obtains comparable performance to baselines that were trained from scratch on the real-world dataset.
>
> > Discuss computational efficiency and scaling behavior
>
>
> Thanks for raising this point! We had provided the computation cost analysis of Cond-FiP versus baselines in the Appendix B.3, but have now moved it to the main text, right at the end of Section 4.
>
> Like other amortized approaches, Cond-FiP has a higher training cost than the baselines, as it is trained across multiple datasets. However, Cond-FiP offers a significant advantage at inference time since it requires only a single forward pass to generate predictions, whereas, the baselines must be retrained from scratch for each new dataset. We find Cond-FiP  offers a $5× speedup$ over FiP in total runtime when evaluating across multiple tasks in our paper.
>
> > Minor textual and formatting revisions
>
> Thanks, we have addressed these concerns as well in the rebuttal revision draft.
>
> We would be happy to provide further clarifications should you have any additional questions. Thank you once again for your time and thoughtful feedback!

---

> > ### Comment · Reviewer_1ybH · 2025-11-12
> >
> > I thank the authors for their thorough and responsive revision. I appreciate the care taken to address each concern raised in my initial review. The additions are well-executed and further strengthen the paper. The paper makes a solid contribution to amortized causal inference and will be of significant interest to the TMLR community. I will therefore be recommending acceptance of the paper.

---

> > > ### Author Response · Authors · 2025-11-17
> > >
> > > Thank you very much for the thoughtful review and for recognizing the contribution of our work. We sincerely appreciate the time you invested and the constructive feedback you provided throughout the process. Your comments greatly helped us strengthen the paper.

---

### Review · Reviewer_EpFg · 2025-10-30

**Summary Of Contributions:**

The authors propose  an amortized inference framework that trains a single transformer-based model to predict the causal mechanisms of Structural Causal Models conditioned on their observational data and causal graph (using Transformer's in-context learning capabilities)

**Audience:**

Yes

**Audience Explanation:**

The method should allow one to train a single model that can easily adapt to new Structural Causal Models, as long as their data obeys the  Additive Noise Model Assumption. Besides the potential applications, the ease of learning and manipulation of causal models is likely an important feature of general intelligence, and this paper shows that for at least some kinds of causal models that is possible.

**Broader Impact Concerns:**

None a priori

**Claims And Evidence:**

Yes

**Claims Explanation:**

The results clearly show the capability of the model to learn SCMs using the datasets and causal graphs, in general matching the traditional Fixed-Point Approach (FiP) and being superior to the other baselines, while being more general than them.

**Requested Changes:**

Some experiments on datasets where the Additive Noise Model Assumption is false to show the impact on performance from a practical perspective (and whether the model can still perform reasonably when ANM is false but an ok approximation). Another important detail missing is a comparison of the computational cost for training and inference of the Cond-FiP model vs the baselines.

---

> ### Author Response · Authors · 2025-11-09
>
> We thank the reviewer for their positive and insightful feedback! We appreciate that they found our results exciting and convincing. We now address the concerns raised by them below.
>
> >  Some experiments on datasets where the Additive Noise Model Assumption is false to show the impact on performance from a practical perspective
>
> We wish to highlight that we also experiment with real-world dataset like flow cytometry (sachs) dataset, and have also added experiments on another real-world dataset (ecoli) in the rebuttal revision. These real-world dataset don't follow additive noise model and our results indicate that Cond-FiP still achieves performance comparable to that of baselines trained from scratch on these datasets.
>
> >  Another important detail missing is a comparison of the computational cost for training and inference of the Cond-FiP model vs the baselines.
>
> Thanks for raising this point! We had provided the computation cost analysis of Cond-FiP versus baselines in the Appendix B.3, but have now moved it to the main text, right at the end of Section 4.
>
> Like other amortized approaches, Cond-FiP has a higher training cost than the baselines, as it is trained across multiple datasets. However, Cond-FiP offers a significant advantage at inference time since it requires only a single forward pass to generate predictions, whereas, the baselines must be retrained from scratch for each new dataset. We find Cond-FiP  offers a $5× speedup$ over FiP in total runtime when evaluating across multiple tasks in our paper.
>
> We would be happy to provide further clarifications should you have any additional questions. Thank you once again for your time and thoughtful feedback!

---

> > ### Comment · Reviewer_EpFg · 2025-11-13
> >
> > I thank the authors for their responses and their revision of the paper. It addresses all the weaknesses I could see in it, and now the benefits of Cond-FiP are even clearer. I'll recommend acceptance of the paper.

---

> > > ### Author Response · Authors · 2025-11-17
> > >
> > > We sincerely appreciate your careful review and constructive feedback. Thank you for acknowledging our revisions and for noting the strengthened clarity around the benefits of Cond-FiP. Your comments greatly helped us strengthen the paper.

---

### Review · Reviewer_n8pZ · 2025-11-04

**Summary Of Contributions:**

The paper proposes an amortized inference framework for predicting the causal mechanisms of a Structural Causal Model (SCM) given observed data and a known causal graph.
Unlike existing methods that require training a separate model for each new dataset, the proposed approach introduces a Transformer-based dataset embedding module, enabling inference of causal mechanisms for unseen graphs and datasets without re-training model parameters.
The method is evaluated on three tasks (noise prediction, data generation, and interventional generation) and achieves performance that is comparable to or better than state-of-the-art baselines in both in-distribution and out-of-distribution (OOD) settings.

**Audience:**

Yes

**Audience Explanation:**

Researchers interested in causal inference, structural causal models (SCMs), and amortized inference would find this work valuable.

**Broader Impact Concerns:**

N/A. No concerns.

**Claims And Evidence:**

Yes

**Claims Explanation:**

The claims of the paper are well supported by comprehensive experiments (by section 4.2))
The authors demonstrate that the proposed amortized inference framework achieves competitive or superior performance to baselines such as FiP, DECI, and DoWhy across multiple benchmarks and data regimes.

---
[In-distribution performance]

Cond-FiP matches or slightly surpasses baselines when both models are evaluated on graphs drawn from the same distribution as training.

---
[Out-of-distribution (OOD) generalization]

The model is evaluated on graphs and data distributions differing from those seen during training including different graph topologies, noise families, and functional parameter shifts. The results show that Cond-FiP maintains stable performance.

---
[Scarce data regimes]

When the number of observational samples is reduced to 50 or 100, Cond-FiP outperforms the baselines trained from scratch.

**Requested Changes:**

I believe the authors’ claims are well supported and the novelty of the work is clear.
None of the following suggestions are critical to secure my recommendation; rather, they would further strengthen the paper.

---
## [Clarifying the benefit of amortization versus additional data]

The proposed model is persuasive, particularly in demonstrating both efficient inference-time performance and competitive predictive performance.

However, it remains somewhat unclear whether the model’s effectiveness truly stems from its ability to capture graph-specific structure, or rather from extensive pretraining over a large number of SCMs.
To strengthen this point, it might be informative to include an experiment that tests the model’s sensitivity to graph structure—for instance, by perturbing the input graph or substituting it with mismatched structures during dataset embedding.
If such perturbations lead to a notable degradation in performance, it would more convincingly demonstrate that the model genuinely relies on graph-specific information rather than benefiting merely from data scale.

---
## [Clarifying the motivation for this specific design]

It might be helpful to elaborate on why this particular amortization structure was chosen over other possible designs. A short conceptual discussion or comparison to plausible alternatives could make the design choice more transparent to readers. Also, it might also be worth discussing whether there exist alternative amortization strategies (beyond FiP-style approaches) that could achieve similar goals.

---
## [Exploring the effect of pretraining scale]

Since the method relies on large-scale pretraining over synthetic SCMs, it could be informative to examine how performance scales with the number of pretraining SCMs (e.g., from 1e5 to 4e6).
Presenting this sensitivity analysis (perhaps even as a brief ablation) would help readers understand how the model’s generalization depends on the scale and diversity of the pretraining corpus.

---

> ### Author Response · Authors · 2025-11-10
>
> We thank the reviewer for their positive and insightful feedback! We appreciate they found our experiments comprehensive and find the empirical evidence compelling. We now address the concerns raised by them below.
>
> > Clarifying the benefit of amortization versus additional data
>
> We thank the reviewer for this great point! Definitely an ablation on perturbing the input causal graph can help us better understand whether Cond-FiP truly learns the causal mechanisms specific to the input causal graph structure.
>
> In the rebuttal revision draft, Section G, we conduct a systematic analysis by introducing random perturbations ($p$) to the true causal graph. Specifically, we randomly remove a proportion $p$ of the true edges, such that on average $p \times$  (total edges) are missing. We report results for sample generation in the challenging OOD setting (d=100 & RFF OUT).
>
> | Method     | p=0      | p=0.01   | p=0.02   | p=0.05   | p=0.1   |
> |-------------|----------|----------|----------|----------|---------|
> | FiP         | 0.55 (0.08) | 0.55 (0.08) | 0.57 (0.08) | 0.62 (0.08) | 0.68 (0.08) |
> | Cond-FiP    | 0.57 (0.07) | 0.58 (0.07) | 0.59 (0.07) | 0.62 (0.07) | 0.67 (0.07) |
>
> Across all tested levels of perturbation, Cond-FiP is competitive with FiP, a baseline trained from scratch for each scenario. These results demonstrate that Cond-FiP exhibits robustness to moderate errors in the input causal graph, comparable to a baseline trained from scratch. This provides more evidence to our claim that Cond-FiP learns causal mechanisms during training and can adapt to new contexts at test time by inferring functions that best explain the available information in the context (input graph and observations), even when the input causal graph is inaccurate.
>
> > Clarifying the motivation for this specific design
>
> The main motivation for adopting the FiP-style amortization design is its ability to represent structural causal models (SCMs) in a flexible, non-parametric manner. Specifically, it models each fixed-point iteration as a forward pass through the (DAG) attention layers, enabling an expressive and data-driven formulation of causal mechanisms without restrictive distributional assumptions.
>
> In contrast, an alternative approach would involve introducing parametric assumptions on the causal mechanisms (e.g., gaussian distribution) and amortizing the estimation of their parameters, as explored by Dhir et al. (2025, arXiv:2507.05526
> ).  The FiP-style design, by avoiding these assumptions, offers a more general formulation that can hopefully better capture complex causal dependencies.
>
> That said, future work could explore hybrid or alternative amortization strategies, such as extending the parametric approach of Dhir et al. (2025) to jointly estimate and sample from the full joint distribution over all causal variables, as our framework currently does. Their present formulation, however, remains limited to modeling distributions over univariate target causal variables.
>
> > Exploring the effect of pretraining scale
>
> Thanks raising this point! We agree that such an analysis would provide further insights into the data requirements of Cond-FiP. Given the limited time and resources, we will try to explore this for few scenarios and report our findings soon.
>
> ---
>
> We would be happy to provide further clarifications should you have any additional questions. Thank you once again for your time and thoughtful feedback!

---

> > ### Comment · Reviewer_n8pZ · 2025-11-16
> >
> > Thank you for providing the additional experiments and clarifying your motivation.
> > Your insights, as well as Section G added to the paper, improved my understanding and have further strengthened the work.
> >
> > Additionally, once the experiment on pretraining scale is completed, I believe it will, as you mentioned, provide valuable information about the data requirements for applying Cond-FiP, further increasing the impact of this paper.
> > I would appreciate it if you could share the results once the experiment is finished!

---

> ### Author Response · Authors · 2025-11-17
>
> We are glad that you found on experiments on benefit of amortization and clarification helpful!
>
> Apologies for the delay, but we have now updated the draft with additional experiments on pretraining scale in Table 22, Appendix H. To better understand the scaling of Cond-FiP w.r.t the pretraining data, we conduct experiments at smaller scales, with a total of $1e5, 4e5, 1e6$ SCMs, as opposed to using $4e6$ SCMs in our main body results.
>
> | Method            | Total Nodes | LIN             | RIN             | LOUT            | ROUT            |
> |------------------|-------------|-----------------|-----------------|-----------------|-----------------|
> | Cond-FiP (1e5)   | 20          | 0.11 (0.01)     | 0.32 (0.06)     | 0.16 (0.03)     | 0.44 (0.04)     |
> | Cond-FiP (4e5)   | 20          | 0.07 (0.01)     | 0.28 (0.06)     | 0.11 (0.01)     | 0.37 (0.03)     |
> | Cond-FiP (1e6)   | 20          | 0.07 (0.01)     | 0.25 (0.06)     | 0.09 (0.01)     | 0.33 (0.03)     |
> | Cond-FiP (4e6)   | 20          | 0.05 (0.01)     | 0.24 (0.06)     | 0.07 (0.01)     | 0.30 (0.03)     |
>
>
> As expected, Cond-FiP benefits consistently from additional pretraining data: all four metrics (\LIN, \RIN, \LOUT, \ROUT) improve monotonically with scale. The most pronounced gains appear when increasing the pretraining size from
> $1e5$ to $4e5$, while improvements become more incremental beyond $1e6$. These results highlight that Cond-FiP continues to leverage larger synthetic datasets, though the returns gradually diminish as scale increases.
>
> Please let us know if you have any additional questions and we remain available to engage further in discussion!

---

### Decision · Action_Editor_keLB · 2025-12-04

**Recommendation:** Accept as is

**Audience:**

Yes

**Audience Explanation:**

The paper lies at the intersection of causal inference, amortized generative modeling, and transformer-based context learning (areas that are highly active within the TMLR community). The idea of building a single model that can generalize across many SCMs and infer mechanisms without retraining is relevant to researchers working on causal modeling, meta-learning, and reusable causal simulators. The framing naturally aligns with current interest in scalable “causal foundation models,” making the work well suited for the TMLR audience.

**Claims And Evidence:**

Yes

**Claims Explanation:**

The paper claims that causal mechanisms of SCMs can be inferred in an amortized fashion using a single model (Cond-FiP), and that this model can perform on par with or better than per-dataset baselines such as FiP, DECI, and DoWhy on noise prediction, observational generation, and interventional generation. These claims are supported by a broad experimental suite including AVICI, C-Suite, Sachs, and ecoli datasets, covering in-distribution, out-of-distribution, scarce-data, graph-size, and simulator-shift settings.
The added rebuttal experiments on graph perturbations and pretraining scale further strengthen the empirical case.
All three reviewers found the evidence convincing and updated their recommendations to clear acceptance.